# A Robust Phased Elimination Algorithm for Corruption-Tolerant Gaussian Process Bandits

**Ilija Bogunovic**
University College London
i.bogunovic@ucl.ac.uk

**Zihan Li**
National University of Singapore
lizihan@u.nus.edu

**Andreas Krause**
ETH Zürich
krausea@ethz.ch

**Jonathan Scarlett**
National University of Singapore
scarlett@comp.nus.edu.sg

## Abstract

We consider the sequential optimization of an unknown, continuous, and expensive to evaluate reward function, from noisy and adversarially corrupted observed rewards. When the corruption attacks are subject to a suitable budget $C$ and the function lives in a Reproducing Kernel Hilbert Space (RKHS), the problem can be posed as *corrupted Gaussian process (GP) bandit optimization*. We propose a novel robust elimination-type algorithm that runs in epochs, combines exploration with infrequent switching to select a small subset of actions, and plays each action for multiple time instants. Our algorithm, *Robust GP Phased Elimination (RGP-PE)*, successfully balances robustness to corruptions with exploration and exploitation such that its performance degrades minimally in the presence (or absence) of adversarial corruptions. When $T$ is the number of samples and $\gamma_T$ is the maximal information gain, the corruption-dependent term in our regret bound is $O(C\gamma_T^{3/2})$, which is significantly tighter than the existing $O(C\sqrt{T\gamma_T})$ for several commonly-considered kernels. We perform the first empirical study of robustness in the corrupted GP bandit setting, and show that our algorithm is robust against a variety of adversarial attacks.

## 1 Introduction

Black-box optimization is a fundamental problem with broad applications including hyperparameter tuning [42], robotics [34], and chemical design [20], among others. To make the problem tractable, a variety of smoothness properties have been adopted, and Reproducing Kernel Hilbert Space (RKHS) functions have proved to provide a versatile framework that can be tackled via Gaussian process (GP) methods [43, 15]. This problem is referred to as *GP bandits* or *kernelized bandits*.

While an extensive line of works have established GP bandit algorithms and regret bounds, settings with adversarial corruptions have only arisen relatively recently. Such corruptions may come in the form of outliers [38, 41], perturbations of sampled inputs [5, 40, 16], adversarial noise in the rewards [8], or perturbations of the final recommendation [7]. In this work, we are interested in the setting of adversarial noise in the rewards, in which the performance of standard non-robust GP bandit algorithms can deteriorate significantly (see Fig. 1).

The first work considering this setting [8] established regret bounds for various algorithms depending on the degree of knowledge on the corruption level $C$ (defined formally in Section 2). A key limitation in their regret bound is that the main corruption-dependent term, $C$, and the usual uncorrupted regret term, which is $\sqrt{T}$ or higher (with time horizon $T$), are *multiplied together*. That is, the dependence

36th Conference on Neural Information Processing Systems (NeurIPS 2022).

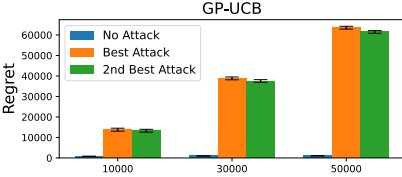
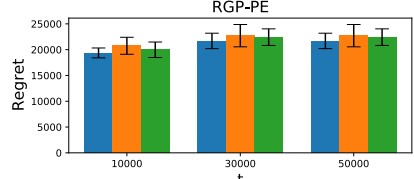

Figure 1: Performance of GP-UCB [43] and Robust GP Phased Elimination (RGP-PE, this work) with no attacks and the two most effective corruption attacks on the Robot3D pushing task. As the number of samples $t$ increases, the performance of non-robust GP-UCB deteriorates significantly under both attacking strategies, while the performance of the proposed algorithm remains robust.

on $C$ is multiplicative with respect to the uncorrupted bound. Analogous studies of bandits with independent arms [35, 21] or linear rewards [9] suggest that *additive* dependence may be possible, but this has remained very much open in the GP bandit setting.

In this paper, we address this fundamental gap in the literature by introducing a novel algorithm in which the uncorrupted term and the $C$-dependent term are clearly decoupled, and the latter is only multiplied by a kernel-dependent function of $T$ that can be much smaller than $\sqrt{T}$.

**Related work.** The closest work to ours is [8], which also considers the corruption-tolerant GP bandit setting. In that work, the authors propose a confidence-bound-based algorithm with enlarged confidence. As outlined above, the regret bound therein scales as $O(C\sqrt{T\gamma_T})$, and the possibility of additive $C$ dependence was left as an open problem.

The question of additive vs. multiplicative dependence first arose in multi-armed bandits with independent arms, with an initial work [35] being multiplicative, and a subsequent work [21] improving to additive. Closer to our setup (and in fact a special case of it via the linear kernel) is the case of corrupted stochastic linear bandits, in which additive dependence was obtained in [9], with the corruption term more precisely being $O(Cd^{3/2}\log T)$ under mild assumptions.[1] Our main result will achieve a similar bound as a special case, while being much more general due to handling general kernels, and adopting GP-based algorithmic and mathematical techniques that have minimal overlap with the linear setting. Other less related results for corrupted linear bandits (e.g., contextual or instance-dependent) are given in [31] and [50].

Adversarial corruptions of the rewards were also considered for GP bandits in [28], but with the key difference of considering a weaker adversary that does not know the chosen action when choosing the corruption term. This distinction has a considerable effect on the problem, leading to significantly different algorithms, and with the setting of [28] leading to a GP-UCB-style regret bound in which the corruptions only impact the constant factors. In our setting, the effect of corruptions is much more significant, and we know from [9] that this is unavoidable in general.

Other less related notions of robustness in GP bandits have included outliers [38], misspecification [13, 6], input noise [5, 40, 16], risk-aversion [39, 11, 37], and corruptions in the final recommendation [7, 29]. Moreover, other settings with adversarial corruptions have included multi-armed bandit and online [21, 25, 24, 3], active [14], reinforcement learning [36, 49, 4], and multi-agent RL [33].

Corruption-robustness have been considered in other sequential decision making problems including multi-armed bandits and predictioin with expert advice / online learning.

**Contributions.** We provide a novel algorithm for GP bandit optimization with adversarial corruptions, that attains the first regret bound to avoid multiplying the uncorrupted part by the corruption level $C$. Our algorithm crucially incorporates a *rare switching* idea, along with a non-standard robust estimator, enlarged confidence bounds, and a minimal number of plays of each selected action; see Sections 2.1 and 3 for details. To our knowledge, we are the first to use rare switching to achieve adversarial robustness; previous works instead used it for reducing computational complexity.

We show that our regret bound is *provably near-optimal* for the SE kernel, and recovers recently-established bounds for stochastic linear bandits [9] that are also known to be near-optimal. For the Matérn kernel, the degree of tightness depends on the dimension and smoothness parameter, but

---

[1] In a paper concurrent with ours, the $Cd^{3/2}$ dependence has been improved to $Cd$ for linear bandits [23]. We leave it for future work to determine whether a similar improvement is possible for GP bandits.

our bound strictly improves on that of [8] in all scaling regimes where the latter is non-trivial (i.e., sub-linear in $T$); see Table 1 on Page 7 for a summary. We demonstrate that our algorithm is able to successfully defend against various attacks, including those proposed in [22].

On the technical side, we note that the GP setting dictates the use of a significantly different algorithm compared to linear bandits, and a technical analysis with only minor overlap. To highlight this, in Appendix E, we explore an approach based on a direct reduction to linear bandits (followed by using the algorithm in [9]), and show that it yields strictly worse regret scaling than our main result.

## 2 Problem Setting and Preliminaries

We consider the Gaussian process bandit (i.e., kernelized bandit) problem, in which the goal of the learner is to maximize the collected rewards by sequentially querying the unknown reward function $f : \mathcal{X} \to \mathbb{R}$ over $T$ rounds. In particular, at every time $t$, the learner selects $x_t \in \mathcal{X}$ and receives

$$y_t = f(x_t) + \epsilon_t, \tag{1}$$

where $\epsilon_t$ is assumed to be $\sigma$-sub-Gaussian with independence over time steps, and $\sigma$ is also known.

We consider the corrupted setting in which, besides the stochastic noise, the observations at every time step are adversarially corrupted, so that the learner observes

$$\widetilde{y}_t = y_t + c_t. \tag{2}$$

Following [8], we make the following assumptions on the adversary:

- The adversary knows the true reward function $f(\cdot)$, and, at every round $t$, it observes $x_t$ before deciding upon the corruption $c_t$.
- The total adversarial corruption budget over $T$ rounds is bounded as follows:

$$\sum_{t=1}^{T} |c_t| \leq C. \tag{3}$$

In this paper, we focus primarily on the case where $C$ is known to the learner, but we also discuss in Section 3.4 how our results have implications for the case of unknown $C$.

The domain $\mathcal{X}$ is assumed to either be finite, or a compact subset of $\mathbb{R}^d$ for some dimension $d$ (e.g., $\mathcal{X} = [0, 1]^d$). In either case, $\mathcal{X}$ is endowed with a continuous, positive semidefinite kernel function $k(\cdot, \cdot) : \mathcal{X} \times \mathcal{X} \to \mathbb{R}$ that is normalized to satisfy $k(x, x') \leq 1$ for all $x, x' \in \mathcal{X}$. We further assume that $f$ has a bounded norm in the corresponding Reproducing Kernel Hilbert Space (RKHS) $\mathcal{H}_k$, i.e., $\|f\|_k \leq B$ (see Appendix A for more details). This assumption permits the construction of confidence bounds via Gaussian process (GP) models (Section 3.2).

The learner's performance is measured using the widely-considered notion of cumulative regret:

$$R_T = \sum_{t=1}^{T} \left( \max_{x \in \mathcal{X}} f(x) - f(x_t) \right), \tag{4}$$

and we are interested in the *joint* dependence of $R_T$ on $C$ and $T$. As noted in [35] and [8], one could alternatively define the cumulative regret with respect to the corrupted values (i.e., $f(x) + c_t$), and these notions coincide to within an additive term of $2C$.

### 2.1 Gaussian Process Model under Corruptions

In the standard (non-corrupted) setting, previous algorithms use (i) zero-mean GP priors for modeling the uncertainty in $f$ (i.e., they assume $f \sim GP(0, k)$), and (ii) Gaussian likelihood models for the observations. As more data points become available, Bayesian posterior updates are then performed according to a misspecified model in which the noise variables $\epsilon_t = y_t - f(x_t)$ are assumed to be drawn independently across $t$ from $\mathcal{N}(0, \lambda)$, where $\lambda$ is a hyperparameter that may differ from the true noise variance $\sigma^2$. In particular, in the absence of corruptions, given a sequence of points $\{x_1, \ldots, x_t\}$ and their noisy observations $\{y_1, \ldots, y_t\}$, the posterior mean and variance are given by

$$\mu_t(x) = k_t(x)^T \big(K_t + \lambda I_t\big)^{-1} Y_t, \tag{5}$$

$$\sigma_t^2(x) = k(x, x) - k_t(x)^T \big(K_t + \lambda I_t\big)^{-1} k_t(x), \tag{6}$$

where $k_t(x) = \left[ k(x_i, x) \right]_{i=1}^{t}$, $K_t = \left[ k(x_t, x_{t'}) \right]_{t,t'}$ is the kernel matrix, and $Y_t \in \mathbb{R}^t$ contains the non-corrupted observations up to time $t$, i.e., $Y_t[i] = y_i$ for $i \in [t]$.

In the corrupted setting, given the inputs $\{x_1, \ldots, x_t\}$ and their corrupted observations $\{\widetilde{y}_1, \ldots, \widetilde{y}_t\}$ (with $\widetilde{y}_i = y_i + c_i$), we propose the following non-standard robust posterior mean estimator:

$$\widetilde{\mu}_t(x) = k_t(x)^T (K_t + \lambda I_t)^{-1} \widetilde{Y}_t, \tag{7}$$

where $\widetilde{Y}_t \in \mathbb{R}^t$ and $\widetilde{Y}_t[i] = \frac{\sum_{j=1}^{t} \mathbb{1}\{x_i = x_j\} \widetilde{y}_j}{\sum_{j=1}^{t} \mathbb{1}\{x_i = x_j\}}$ for $i \in [t]$. Intuitively, the averaging of terms corresponding to identical actions is done in order to diminish the impact of corruption, and this will be a crucial component of our analysis. In our algorithm, besides $\widetilde{\mu}_t(\cdot)$, we will also make use of the standard posterior variance $\sigma_t^2(\cdot)$ as given in Eq. (6); the use of this quantity is intuitively reasonable because GP posterior variances do not depend on the observations.

The main quantity that characterizes the regret bounds in the non-corrupted setting (and is also useful in our setting) is the *maximum information gain* [43], defined at time $t$ as

$$\gamma_t = \max_{x_1, \ldots, x_t} \frac{1}{2} \ln \det(I_t + \lambda^{-1} K_t). \tag{8}$$

# 3 Robust GP Phased Elimination

## 3.1 Algorithm and Confidence Bounds

Our algorithm works in epochs indexed by $h = 0, 1, \ldots, H - 1$, each of which consists of sampling a batch of points. The epoch lengths may be chosen adaptively, and hence $H$ may not be deterministic, but we will ensure with probability one that $H \leq \bar{H}$ with $\bar{H} = \log_2 T$. The length of epoch $h$ is denoted by $u_h$, so that $\sum_{h=0}^{H-1} u_h = T$.

The algorithm and analysis are based on the widespread notion of confidence bounds. While our confidence bounds will be expanded to account for corruptions, it is useful to consider the following generic assumption regarding non-corrupted observations (although the algorithm cannot access these, they will appear in our mathematical analysis).

**Assumption 1** (Regular confidence bounds). *Let $\mu^{(h)}(x)$ and $\sigma^{(h)}(x)$ denote the posterior mean and standard deviation computed (hypothetically) using only the non-corrupted observations $\{(x_i, y_i)\}_{i=1}^{u_h}$ in epoch $h$ using Eqs. (5) and (6). We assume that given $\delta \in (0, 1)$, there exists a sequence of parameters $\beta_h = \beta_h(\delta)$ which is non-decreasing in $h$ and yields with probability at least $1 - \delta$ that*

$$|\mu^{(h)}(x) - f(x)| \leq \beta_h \sigma^{(h)}(x) \tag{9}$$

*simultaneously for all $h \geq 0$ and $x \in \mathcal{X}$.*

Specific choices of $\beta_h$ satisfying this assumption will be considered in Section 3.2.

Similarly to previous kernelized algorithms (e.g., [8, 6]), our proposed algorithm makes use of enlarged confidence bounds. Hence, our first result concerns concentration of an RKHS member under corrupted observations, where we make use of the proposed estimator from Eq. (7).

**Lemma 2** (Corrupted confidence bounds). *Under Assumption 1, let $\widetilde{\mu}^{(h)}(x)$ denote the posterior mean based on only the corrupted observations $\{(x_i, \widetilde{y}_i)\}_{i=1}^{u_h}$ in epoch $h$ using Eq. (7), and let $u_{\min} \geq 1$ denote the minimum number of times any single action from $\{x_i\}_{i=1}^{u_h}$ is played, i.e., $u_{\min} = \min_{x \in \{x_1, \ldots, x_{u_h}\}} \sum_{i=1}^{u_h} \mathbb{1}\{x_i = x\}$. Then, with probability at least $1 - \delta$, it holds for all $x \in \mathcal{X}$ and $h \geq 0$ that*

$$|\widetilde{\mu}^{(h)}(x) - f(x)| \leq \left( \beta_h + \frac{C\sqrt{u_h}}{u_{\min}\lambda} \right) \sigma^{(h)}(x). \tag{10}$$

The confidence-bound enlargement is proportional to the total amount of corruption $C$. This bears some similarity to the confidence intervals used in [8, Lemma 2], but we note the following important differences:

- We make use of a novel kernelized mean estimator (Eq. (7)) that takes average over rewards corresponding to the same played action;

---

**Algorithm 1** Robust GP Phased Elimination (RGP-PE)

---

**Input:** Domain $\mathcal{X} \subset \mathbb{R}^d$, truncation parameter $\psi > 0$, corruption budget $C$, switching parameter $\eta > 1$, regularization parameter $\lambda > 0$

1: Initialize $l_0 = 2$, and $h = 0$ and $\mathcal{X}_h = \mathcal{X}$
2: Set $\mathcal{S}_h = \emptyset$, $t' = 0$, $\sigma_0(x) = 1$ for all $x \in \mathcal{X}_h$
3: **for** $t = 1, 2, \ldots, l_h$ **do**
4:     Select $x_t = \arg\max_{x \in \mathcal{X}_h} \sigma_{t'}(x)$
5:     Update $\mathcal{S}_h \leftarrow \mathcal{S}_h \cup \{x_t\}$
6:     **if** $\det(I_t + \lambda^{-1} K_t) > \eta \det(I_{t'} + \lambda^{-1} K_{t'})$ **then**
7:         Set $t' \leftarrow t$
8:         Compute $\sigma_{t'}(\cdot)$ via Eq. (6) by using $\{x_i\}_{i=1}^{t'}$
9:     **end if**
10: **end for**
11: Set $\xi_h(x) = \frac{\sum_{i=1}^{l_h} \mathbb{1}\{x = x_i\}}{l_h}$ for every $x \in \mathcal{S}_h$
12: Set $u_h(x) = \lceil l_h \max\{\xi_h(x), \psi\} \rceil$ for every $x \in \mathcal{S}_h$
13: Take each action $x \in \mathcal{S}_h$ exactly $u_h(x)$ times with corresponding rewards $(\widetilde{y}_j)_{j=1}^{u_h}$ where $u_h = \sum_{x \in \mathcal{S}_h} u_h(x)$
14: Estimate $\widetilde{\mu}^{(h)}(\cdot)$ and $\sigma^{(h)}(\cdot)$ according to Eq. (7) and Eq. (6) using only the $u_h$ points from the current epoch.
15: Update the active set of actions to:

$$\mathcal{X}_{h+1} \leftarrow \Big\{ x \in \mathcal{X}_h : \widetilde{\mu}^{(h)}(x) + \big(\beta_h + \tfrac{C\sqrt{u_h}}{l_h \psi \lambda}\big)\sigma^{(h)}(x) \geq$$
$$\max_{x \in \mathcal{X}_h} \widetilde{\mu}^{(h)}(x) - \big(\beta_h + \tfrac{C\sqrt{u_h}}{l_h \psi \lambda}\big)\sigma^{(h)}(x) \Big\}$$

16: Set $l_{h+1} \leftarrow 2l_h$, $h \leftarrow h + 1$ and return to Step 2 (terminating after $T$ total actions are played).

---

- Our enlargement term is $O(C \frac{\sqrt{u_h}}{u_{\min}})$, as opposed to $O(C)$ used in [8, Lemma 2]. We will typically apply this lemma with $\frac{\sqrt{u_h}}{u_{\min}} \ll 1$, so that our confidence width is much smaller.

For the second of these, the intuition is that if the same action is played multiple times, it becomes harder for the adversary to hide the true value (i.e., since the rewards of the same played actions are averaged, the adversary needs to spend more of its budget corrupting the reward).

The Robust GP-Phased Elimination algorithm (Algorithm 1) proceeds in epochs (indexed by $h$) of exponentially increasing length $u_h$. At every round $t$ (where $t \in \{1, \ldots, l_h\}$ and $l_h = 2^{h+1}$) within an epoch $h$, the algorithm selects an action maximizing a posterior uncertainty computed at some (possibly strictly earlier) time $t'$:

$$x_t = \arg\max_{x \in \mathcal{X}_h} \sigma_{t'}(x), \tag{11}$$

where $\mathcal{X}_h$ denotes the set of active actions in epoch $h$. The selected action is then added to $\mathcal{S}_h$ which is a set that contains distinct actions selected in epoch $h$.

The key idea behind using $t'$ instead of $t$ in Eq. (11) is to ensure that our algorithm *rarely switches*, based on a condition relating to the information gain (Line 6), meaning that the same action $x_t$ is typically selected multiple times. Whenever there are ties, they are resolved arbitrarily but consistently over rounds (i.e., if $\sigma_{t'}(\cdot)$ does not change, the same points are selected). Based on Lines 6 to 9, we update $t'$ and recompute $\sigma_{t'}(x)$ only when $\det(I_t + \lambda^{-1} K_t)$ increases by a constant factor $\eta$.

Related ideas of rare switching have appeared in the literature (e.g., [1, 47, 19]), but to our knowledge we are the first to use this idea in the kernelized bandit problem to provide an algorithm that includes an explicit switching condition for improving robustness. Intuitively, by rarely switching, we obtain more samples of the same point, allowing us to average more of them together and making the "averaged" observation harder to corrupt. Concurrent work also used rare switching to reduce GP posterior computation, noting that the computation time can be made to scale (cubically) with the number of *unique* points [12]. This benefit also applies directly to our algorithm, and we exploit it to run large-$T$ experiments in Section 4.

After the set $\mathcal{S}_h$ is constructed, we define $\xi_h(x) = \frac{\sum_{i=1}^{l_h} \mathbb{1}\{x = x_i\}}{l_h}$ for every $x \in \mathcal{S}_h$, representing the empirical frequency of selecting $x_t \in \mathcal{X}_h$ in $l_h$ rounds. The algorithm then plays actions from $\mathcal{S}_h$ only, where the number of times each action $x$ from $\mathcal{S}_h$ is played is denoted by $u_h(x) = \lceil l_h \max\{\xi_h(x), \psi\} \rceil$. Here, the *truncation parameter $\psi$* ensures that each action from $\mathcal{S}_h$ is played sufficiently many times; this idea was used for corrupted linear bandits in [9]. Our theory suggests a particular choice of $\psi$; see Theorem 3. Each action $x \in \mathcal{S}_h$ is played for $u_h(x)$ times in an arbitrary order, leading to the total epoch length $u_h = \sum_{x \in \mathcal{S}_h} u_h(x)$.

Based on the received noisy and potentially corrupted rewards $\{x_j, \widetilde{y}_j\}_{j=1}^{u_h}$, the algorithm updates its estimates $\widetilde{\mu}^{(h)}(\cdot)$ and $\sigma^{(h)}(\cdot)$ according to Eq. (7) and Eq. (6). Finally, each epoch $h$ ends by updating the set of active actions $\mathcal{X}_{h+1}$. To do so, we use the confidence bounds from Lemma 2 with $u_{\min} = l_h \psi$, where $l_h \psi$ is a lower bound on the number of times each distinct action from $\mathcal{S}_h$ is played. These confidence bounds are valid in the sense that the true function is contained within the confidence bounds with high probability. The definition of $\mathcal{X}_{h+1}$ (Line 15) ensures that with high probability, the optimal action is never eliminated.

Besides the standard exploration/exploitation trade-off (controlled via $\beta_h$), our algorithm additionally balances robustness to corruptions. This is done via two parameters: the switching parameter $\eta$ and truncation parameter $\psi$. We set these parameter to ensure that the number of distinct actions played per epoch is sufficiently small, while the number of plays per each such action is sufficiently large. This trade-off is non-trivial; for example, in the case that $C = 0$ (i.e., the non-corrupted setting), resampling the same actions (controlled via $\psi$) increases the regret.

**Main result.** We now present our main theoretical result, where we use $O^*(\cdot)$ notation to hide constants and dimension-independent log factors. We treat the RKHS norm bound $B$ as being fixed, so its dependence is also hidden in $O(\cdot)$ or $O^*(\cdot)$ notation.

**Theorem 3** (Main result). *Under the preceding setup and Assumption 1, for any corruption budget $C \geq 0$, Algorithm 1 with a constant switching parameter $\eta > 1$ and truncation parameter $\psi = \frac{\ln \eta}{2\gamma_T}$ satisfies the following with probability at least $1 - \delta$:*

$$R_T = O^*\big(\beta_{\bar{H}} \sqrt{T \gamma_T} + C \gamma_T^{3/2}\big). \tag{12}$$

### 3.2 Applications to Specific Confidence Bounds

Now we discuss specific choices of $\beta_h$ satisfying Assumption 1, and the resulting final regret bounds.

We observe that the actions in each fixed epoch are sampled non-adaptively, and the resulting GP posterior formed only depends on the points in that epoch. As noted in [32], these conditions are sufficient to make use of the following confidence bounds for non-adaptive sampling.

**Lemma 4.** [45, Theorem 1] *When $\{x_i\}_{i=1}^t$ are selected independently of all the observations $\{y_i\}_{i=1}^t$, it holds for any fixed $x \in \mathcal{X}$ and any $t \geq 1$ with probability at least $1 - \delta$ that $|\mu_t(x) - f(x)| \leq \big(B + \frac{\sigma}{\sqrt{\lambda}} \sqrt{2 \log \frac{1}{\delta}}\big) \sigma_t(x).$*

For finite domains, applying the union bound leads to a choice of $\beta_h$ for the proposed algorithm such that $\beta_{\bar{H}}$ only contributes to logarithmic terms in the cumulative regret.

**Corollary 5.** *Defining $\bar{\beta}_h(\delta) = B + \frac{\sigma}{\sqrt{\lambda}} \sqrt{2 \log \frac{|\mathcal{X}|}{\delta}}$, we have that Assumption 1 holds with $\beta_h = \bar{\beta}_h(\delta_h)$ and $\delta_h = \frac{6\delta}{(h+1)^2 \pi^2}$. Hence, with probability at least $1 - \delta$, Algorithm 1 with switching parameter $\eta > 1$, truncation parameter $\psi = \frac{\ln \eta}{2\gamma_T}$, and $\beta_h$ as above achieves*

$$R_T = O^*\big(\sqrt{T \gamma_T} + C \gamma_T^{3/2}\big). \tag{13}$$

This corollary is obtained by noting that the error probability is at most $\delta$ as desired, since a union bound over $\mathcal{X}$ gives a per-epoch term of at most $\delta_h$, and $\sum_{h=0}^{H-1} \delta_h \leq \sum_{h=0}^{\infty} \frac{6\delta}{(h+1)^2 \pi^2} = \big(\sum_{h=0}^{\infty} \frac{1}{(h+1)^2}\big) \frac{6\delta}{\pi^2} \leq \frac{\pi^2}{6} \cdot \frac{6\delta}{\pi^2} = \delta.$

For general (possibly continuous) domains, one option is to set $\beta_h$ according to a widely-used confidence bound as follows, though we will shortly discuss improved choices.

| Kernel | Lower Bound | Existing | Ours |
|--------|-------------|----------|------|
| Linear | $\sqrt{Td} + Cd$ | $\sqrt{Td} + Cd^{3/2}$ | $\sqrt{Td} + Cd^{3/2}$ |
| SE | $\sqrt{T}(\log T)^{d/2} + C(\log T)^{d/2}$ | $\sqrt{T}(\log T)^d + C\sqrt{T}(\log T)^{d/2}$ | $\sqrt{T}(\log T)^d + C(\log T)^{3d/2}$ |
| Matérn | $T^{\frac{\nu+d}{2\nu+d}} + C^{\frac{\nu}{d+\nu}}T^{\frac{d}{d+\nu}}$ | $T^{\frac{2\nu+3d}{4\nu+2d}} + CT^{\frac{\nu+d}{2\nu+d}}$ | $T^{\frac{\nu+d}{2\nu+d}} + CT^{\frac{3d}{4\nu+2d}}$ |

Table 1: Summary of regret bounds with constants and dimension-independent log factors omitted. For the SE and Matérn kernels, the upper bounds are from [8] and the lower bounds are from [10]. For the linear kernel, the existing bounds are from [9], except the $\sqrt{Td}$ lower bound which is from [17].

**Lemma 6.** [15, Theorem 2] *For any (possibly adaptive) sampling strategy, it holds with probability at least $1 - \delta$ that $|\mu_t(x) - f(x)| \leq \left(B + \sigma\sqrt{2(\gamma_t + 1 + \ln(1/\delta))}\right)\sigma_t(x)$ for all $x \in \mathcal{X}$ and $t \geq 1$.*

By a similar argument to Corollary 5 and the fact that $\gamma_t$ is increasing in $t$, we obtain the following.

**Corollary 7.** *If $u_h \leq \bar{u}_h$ almost surely, then defining $\check{\beta}_h(\delta) = B + \sigma\sqrt{2(\gamma_{\bar{u}_h} + 1 + \ln(1/\delta))}$, we have that Assumption 1 holds with $\beta_h = \check{\beta}_h(\delta_h)$ and $\delta_h = \frac{6\delta}{(h+1)^2\pi^2}$. Hence, with probability at least $1 - \delta$, Algorithm 1 with a constant switching parameter $\eta > 1$, truncation parameter $\psi = \frac{\ln \eta}{2\gamma_T}$, and $\beta_h$ as above achieves*

$$R_T = O^*\left(\sqrt{T}\gamma_T + C\gamma_T^{3/2}\right), \tag{14}$$

*where we crudely selected $\bar{u}_h = T$.*

While this regret bound can be significantly weaker than Corollary 5 due to the $O^*(\sqrt{T}\gamma_T)$ term, we can also obtain an analog of Corollary 5 (i.e., attaining the improved dependence in Eq. (13)) for continuous domains, under the mild assumption that functions in the RKHS are Lipschitz continuous (which is true for the kernels we consider below). A crude approach is to have the algorithm use a very fine discretization [26, 32], and a more sophisticated approach is to only discretize as part of the analysis [45]. The details can be found in the preceding references, and we avoid repeating them.

### 3.3 Comparisons to Existing Bounds

We specialize our regret bound in Eq. (13) to specific kernels by substituting $\gamma_T = O^*(d)$ for the linear kernel, $\gamma_T = O^*((\log T)^d)$ for the SE kernel, and $\gamma_T = O^*(T^{\frac{d}{2\nu+d}})$ for the Matérn kernel [43]. The resulting regret bounds are shown in Table 1 (omitting constants and dimension-independent log factors), along with the best known existing upper and lower bounds. We observe the following:

- For the linear kernel, we recover the recent upper bound of [9], and this is tight up to the presence of $d$ vs. $d^{3/2}$ in the corrupted part.
- For the SE kernel, we match the lower bound of [10] up to small changes in the implied constant in each $(\log T)^{\Theta(d)}$ term. In contrast, the existing upper bound of [8] incurs a much larger $\sqrt{T}$ term in the corrupted part.
- For the Matérn kernel, compared to the existing result in [8], we obtain an improvement in the non-corrupted part recently established in [32], matching the non-corrupted lower bound. In the corrupted part, the existing result has a better exponent to $T$ when $\nu < \frac{d}{2}$, whereas ours is better when $\nu > \frac{d}{2}$, in particular approaching zero (instead of $\frac{1}{2}$) as $\nu \to \infty$ and nearly matching the lower bound in this limit. However, when $\nu < \frac{d}{2}$ we find that the non-corrupted part in [8] is super-linear in $T$, making the bound trivial. Hence, our bound is better whenever non-trivial scaling is attained.

The bounds based on a reduction to linear bandits, which we derive in Appendix E, are omitted in Table 1. We briefly note that they are able to provide a similar upper bound to our main one under the SE kernel, but are always strictly worse under the Matérn kernel.

### 3.4 Implications for the Unknown $C$ Setting

While we have focused on the case of known $C$, an idea from a concurrent work [23] (on linear bandits) can be used to transfer our main result to a setting with unknown $C$.

The idea is that if the parameter $C$ is used by the algorithm but $C_{\text{true}}$ is the amount of corruption actually used by the adversary, then the analysis goes through unchanged as long as $C \geq C_{\text{true}}$.

Hence, we may cautiously choose a large value of $C$ to cover more values of $C_{\text{true}}$. As an important special case, we may choose $C$ such that the corrupted and uncorrupted regret terms are of the same order; for instance, in (13), setting $C = O\left(\frac{\sqrt{T}}{\gamma_T}\right)$ gives $R_T = O^*(\sqrt{T\gamma_T})$. Hence, we find that any corruption level $C_{\text{true}}$ up to $O\left(\frac{\sqrt{T}}{\gamma_T}\right)$ only affects the constant (or possibly logarithmic) factors, and the precise corruption level does not need to be known.

For particularly smooth kernels such as linear and SE (with constant dimension), the scaling $O\left(\frac{\sqrt{T}}{\gamma_T}\right)$ reduces to $O^*(\sqrt{T})$. This may not seem as high as ideal, but at least in the case of linear bandits, it is known to be the best we can hope for unless the algorithm attains significantly higher uncorrupted regret [9, 23]. Specifically, if optimal $O^*(\sqrt{T})$ uncorrupted regret is attained, then linear regret is unavoidable when $C = \omega(\sqrt{T})$. See [23] for similar statements with the dependence on $d$ included.

The overall picture remains less complete for general kernels, but the preceding discussion reveals that our results for known $C$ do have important implications for the unknown $C$ setting.

## 4 Experiments

We experimentally evaluate the performance of our proposed algorithm, along with two baselines, one robust and one non-robust. Our experiments serve as a proof of concept for our proposed approach, but also highlight possible remaining gaps between theory and practice, e.g., arising from large constant factors in the regret bounds. We emphasize that our contributions are primarily theoretical.

**Algorithms.** We consider the following three algorithms:

1. RGP-PE: Robust GP-Phased Elimination with constant $\beta_h$; this is a slight variation of Corollary 5 in which the number of epochs $H$ turns out to be a small constant in our experiments.
2. GP-UCB: a representative non-robust fully sequential algorithm with slowly growing $\beta_t$, where $t \in [T]$ [43, Algorithm 1].
3. RGP-UCB: the robust version of GP-UCB with slowly growing $\beta_t$ [8, Algorithm 1], where the only difference from GP-UCB is that the theoretical coefficient of $\sigma_{t-1}$ in the UCB is $\beta_t + \frac{C}{\sqrt{\lambda}}$.

We found the term $\beta_h + \frac{C\sqrt{u_h}}{l_h\psi\lambda}$ multiplying $\sigma^{(h)}$ in Algorithm 1 to be overly conservative, so we instead replace it by $\beta_h + b \cdot \frac{C}{\sqrt{u_h}}$ (since $l_h$ and $u_h$ are similar, we replace $\frac{\sqrt{u_h}}{l_h}$ by $\frac{1}{\sqrt{u_h}}$), where $b \in (0,1]$ is an additional parameter controlling the degree of exploration and robustness. Similarly, in RGP-UCB we use the coefficient $\beta_t + b \cdot \frac{C}{\sqrt{\lambda}}$. The remaining parameters $\beta_h$ and $\beta_t$ are specified below.

**Synthetic Function.** We produce a synthetic 2D function $f_1$, shown in Figure 4 of the supplementary material, which is randomly sampled from a Gaussian Process with zero mean and the SE kernel with lengthscale $l = 0.5$. The domain $\mathcal{X}$ of $f_1$ contains 100 points obtained by evenly splitting $[-5,5]^2$ into a $10 \times 10$ grid. We use the true kernel as the prior for all three algorithms, and use $\beta_h = 4$ for RGP-PE, and $\beta_t = \sqrt{\log t}/2$ for GP-UCB and RGP-UCB.

**Robot Pushing Objective Function.** We consider the deterministic robot pushing objective function on a 2D plane introduced in [48], which aims to find suitable parameters to push an object to the target location $r_g$. We use the Robot3d function, which takes the robot location $(r_x, r_y)$ and pushing duration $t_r$ as a 3D input, and outputs the reversed distance between the pushed robot location and the target location $r_g$, i.e.,

$$\text{Robot3D}(r_x, r_y, t_r) = 5 - \|\text{push}(r_x, r_y, t_r) - r_g\|,$$

where $\text{push}(\cdot)$ outputs the pushed robot location.

We let the domain $\mathcal{X}$ contain 100 points $(r_x, r_y, t_r)$ randomly sampled from $[-5,5]^2 \times [1,30]$, and the target location $r_g$ is set to be $(3,2)$. Since the lengthscale of the SE kernel with maximum likelihood given the noiseless data is $1.94 \approx 2$, we use the SE kernel with $l = 2$ as prior for all three algorithms. We found it beneficial for all algorithms to be slightly more explorative for this function, and accordingly use $\beta_h = 6$ for KE and $\beta_t = 2\sqrt{\log t}$ for GP-UCB and RGP-UCB.

**Attack Methods.** We consider the following five attack methods, which continue until the corruption budget is exhausted:

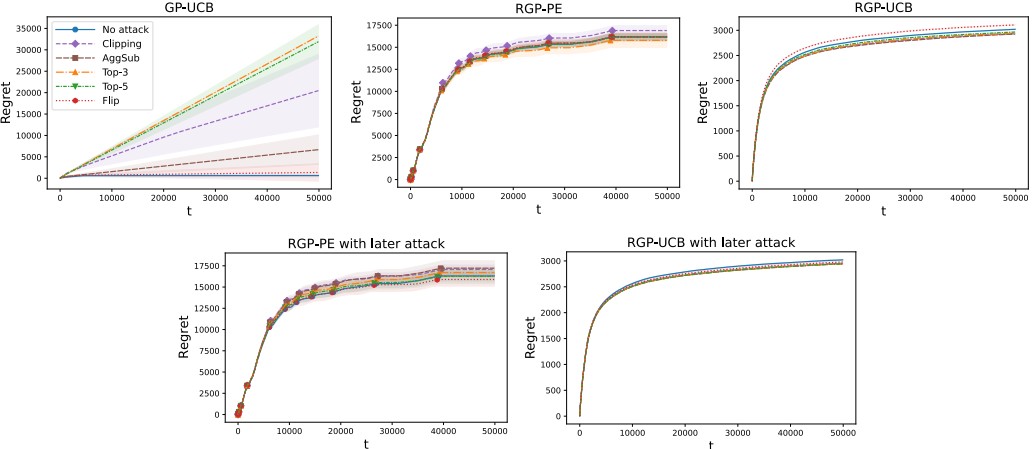

Figure 2: Performance on $f_1$ with $C = 50$. We observe that GP-UCB incurs linear regret for several attacks, whereas the other algorithms exhibit robustness to all of the attacks.

- Clipping: This attack proposed in [22] perturbs $f$ and produces another reward function $\widetilde{f}$ whose optima are in some region $\mathcal{R}_{\text{target}}$ that does not contain $x^*$ by setting

$$\widetilde{f}(x) = \begin{cases} f(x) & x \in \mathcal{R}_{\text{target}}, \\ \min\{f(x), f(\widetilde{x}^*) - \Delta\} & x \notin \mathcal{R}_{\text{target}}, \end{cases}$$

  where $\widetilde{x}^* = \arg\max_{x \in \mathcal{R}_{\text{target}}} f(x)$. We let $\Delta = 0.5$ and choose $\mathcal{R}_{\text{target}} = \{(x_1, x_2) \in \mathcal{X} : x_1 \leq x_2\}$ for $f_1$, and $\mathcal{R}_{\text{target}} = \{(r_x, r_y, t_r) \in \mathcal{X} : r_x \geq 0\}$ for the function Robot3D.

- Aggressive Subtraction (AggSub): This attack proposed in [22] sets

$$\widetilde{f}(x) = \begin{cases} f(x) & x \in \mathcal{R}_{\text{target}}, \\ f(x) - h_{\max} & x \notin \mathcal{R}_{\text{target}}, \end{cases}$$

  for some $h_{\max} > f(x^*) - f(\widetilde{x}^*)$. We use the same $\mathcal{R}_{\text{target}}$ as the Clipping attack, and let $h_{\max} = 1$ for $f_1$ and $h_{\max} = 3$ for Robot3D.

- Top-$K$: When $x$ is one of the top $K$ remaining actions, this attack perturbs the reward down to $-1$. We consider both $K = 3$ and $K = 5$.

- Flip: This attack simply flips the reward from $f(x)$ to $-f(x)$. Both this attack and the previous one are variations of attacks considered for linear bandits in [9].

For the algorithms, we consider $C = 50$ and $C = 100$. By default, the attack starts at $t = 1$, but for the robust algorithms RGP-PE and RGP-UCB, we also conduct experiments with a *later* attack, where (i) the attack in RGP-PE starts when at least one action is eliminated from the domain; and (ii) the attack in RGP-UCB starts when at least one action has UCB strictly lower than $\max_{x \in \mathcal{X}} \text{LCB}(x)$.

We let $T = 50000$,[2] $\sigma = 0.02$, and $\lambda = 1$ for all three algorithms, $b = 0.1$ for RGP-PE and RGP-UCB, and $\psi = 0.5, \eta = 2$ for RGP-PE. The results are produced by performing 10 trials and plotting the average cumulative regret, with error bars indicating one standard deviation.

**Comparison of Algorithms.** As shown in Figures 2 and 3, the non-robust algorithm GP-UCB succeeds when no attack is applied. However, the cumulative regret for $f_1$ associated with the Clipping, AggSub, Top-3, and Top-5 attacks grow linearly, indicating that these four attacks succeed in driving GP-UCB towards a suboptimal action. Similarly, the Top-3 and Top-5 attacks incur linear regret for Robot3D. In contrast, we find that RGP-PE has only one action remaining at the end of the 13th epoch, and manages to defend against all five attack methods for both functions.

The baseline robust algorithm RGP-UCB also successfully defends against all the attacks, and generally has lower cumulative regret than RGP-PE, despite RGP-PE having a stronger regret

---

[2]As we mentioned previously, this large value of $T$ is feasible due to the computation time only scaling with respect to the number of *unique* points [12], which is much smaller than $T$.

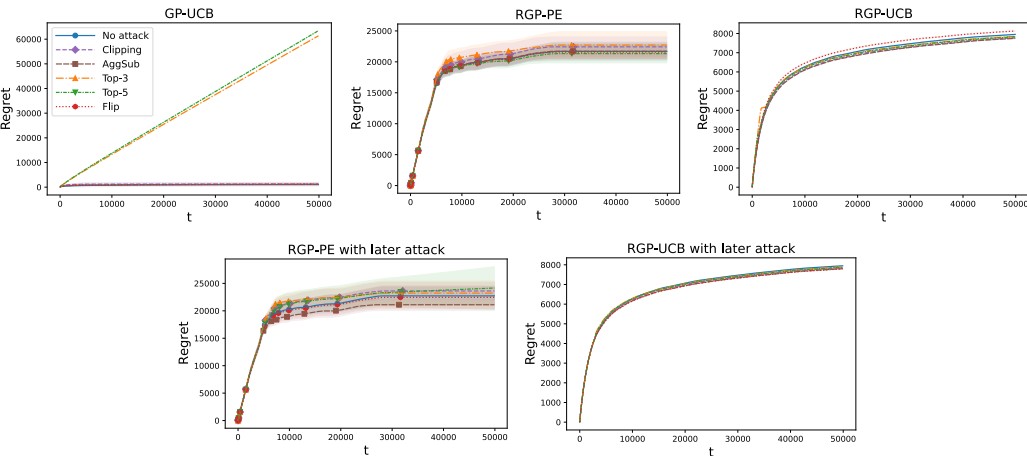

Figure 3: Performance on Robot3D with $C = 100$. We observe that GP-UCB incurs linear regret for two attacks, whereas the other algorithms exhibit robustness to all of the attacks.

guarantee. There are at least two possibly reasons for this: (i) The analysis of RGP-UCB in [8] could be loose, with a tighter analysis potentially giving an additive dependence similar to Theorem 3, and (ii) the strong scaling laws in our theory may still leave room for improvements in the constant factors (or logarithmic). Further addressing these findings remains an interesting direction for future work. We note that even in the more specialized problem of corrupted stochastic linear bandits, analogous practical limitations of a phased elimination algorithm were observed in [9].

**Later Attack.** We observe that RGP-PE and RGP-UCB are also able to defend against the later attack, and their performance is similar to when the attack starts from the beginning. There are only two trials of RGP-PE (budget $C = 100$ and Top-5 attack on Robot3D in Figure 3), in which the only action remaining at the end of the 13th epoch is slightly suboptimal. In Appendix F, we additionally show the experiment results for $f_1$ with $C = 100$, and Robot3D with $C = 50$.

## 5   Conclusion

We have provided a new algorithm for corruption-tolerant GP bandits based on phased elimination, incorporating a key idea of *rare switching* based on a certain condition relating to the information gain, along with a robust estimator, enlarged confidence bounds, and truncation to ensure a minimal number of plays of each selected action. Our regret bound recovers the best known existing bound under the linear kernel, is provably near-optimal under the SE kernel, and improves on the best existing bound in all cases where the latter is non-trivial.

## Acknowledgment

This project has received funding from the European Research Council (ERC) under the European Unions Horizon 2020 research and innovation programme grant agreement No 815943. J. Scarlett was supported by the Singapore National Research Foundation (NRF) under grant number R-252-000-A74-281.

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
