**(Bogunovic/Li/Krause/Scarlett, NeurIPS 2022)**

## A   Preliminaries

Here, we outline some useful and well-known results and definitions typically used in kernelized/GP bandit (Bayesian optimization) algorithms.

**RKHS and kernel functions.** We denote by $\mathcal{H}_k$ the reproducing kernel Hilbert space (RKHS) corresponding to the kernel $k$, defined as a Hilbert space of functions equipped with an inner product $\langle \cdot, \cdot \rangle_k$, satisfying the reproducing property, i.e., $\langle f(\cdot), k(\cdot, x) \rangle_k = f(x), \forall x \in \mathcal{X}, \forall f \in \mathcal{H}_k$.

Since we assume that the kernel is bounded (i.e., $k(x, x') \leq 1$), continuous, and has a compact domain (namely, $D = [0,1]^d$), the conditions of Mercer's theorem are satisfied [27], and the kernel admits a countably infinite (or finite) dimensional feature space, i.e., there exists $\{(\lambda_m, \phi_m)\}_{m=1}^{\infty}$ such that $k(x, x') = \sum_{m=1}^{\infty} \lambda_m \phi_m(x) \phi_m(x')$ where the $\phi_m(\cdot)$ are eigenfunctions, and the $\lambda_m \geq 0$ are eigenvalues. We form an infinite-dimensional feature vector as follows:

$$\phi(x) = (\sqrt{\lambda_1}\phi_1(x), \sqrt{\lambda_2}\phi_2(x), \dots), \tag{15}$$

which yields $k(x, x') = \phi(x)^T \phi(x')$. As stated in the main text, we assume that the RKHS norm is upper bounded by some constant $B > 0$.

The following lemma provides a useful expression for $\sigma_t^2(x)$. This result is fairly standard, but for completeness, we provide a short proof. Here and subsequently, we use $I$ to denote the infinite-dimensional identity matrix in feature space.

**Lemma 8.** *Defining* $\Phi_t = [\phi(x_1), \dots, \phi(x_t)]^T$, *we have*

$$\sigma_t^2(x) = \lambda \phi(x)^T (\Phi_t^T \Phi_t + \lambda I)^{-1} \phi(x). \tag{16}$$

*Proof.* We can rewrite $\sigma_t^2(x)$ as follows,

$$\sigma_t^2(x) = k(x, x) - k_t(x)^T (K_t + \lambda I_t)^{-1} k_t(x) \tag{17}$$

$$= \phi(x)^T \phi(x) - \phi(x)^T \Phi_t^T (\Phi_t \Phi_t^T + \lambda I_t)^{-1} \Phi_t \phi(x) \tag{18}$$

$$= \phi(x)^T \Phi_t^T (\Phi_t \Phi_t^T + \lambda I_t)^{-1} \Phi_t \phi(x) + \lambda \phi(x)^T (\Phi_t^T \Phi_t + \lambda I)^{-1} \phi(x)$$
$$\qquad - \phi(x)^T \Phi_t^T (\Phi_t \Phi_t^T + \lambda I_t)^{-1} \Phi_t \phi(x) \tag{19}$$

$$= \lambda \phi(x)^T (\Phi_t^T \Phi_t + \lambda I)^{-1} \phi(x), \tag{20}$$

where Eq. (19) uses $\phi(x) = \Phi_t^T (\Phi_t \Phi_t^T + \lambda I_t)^{-1} \Phi_t \phi(x) + \lambda (\Phi_t^T \Phi_t + \lambda I)^{-1} \phi(x)$, which can be obtained as follows.

$$(\Phi_t^T \Phi_t + \lambda I)\phi(x) = \Phi_t^T \Phi_t \phi(x) + \lambda \phi(x) \tag{21}$$

$$\phi(x) = (\Phi_t^T \Phi_t + \lambda I)^{-1} \Phi_t^T \Phi_t \phi(x) + \lambda (\Phi_t^T \Phi_t + \lambda I)^{-1} \phi(x) \tag{22}$$

$$= \Phi_t^T (\Phi_t \Phi_t^T + \lambda I_t)^{-1} \Phi_t \phi(x) + \lambda (\Phi_t^T \Phi_t + \lambda I)^{-1} \phi(x), \tag{23}$$

where the last step follows from the standard push-through identity $(\Phi_t^T \Phi_t + \lambda I)\Phi_t^T = \Phi_t^T (\Phi_t \Phi_t^T + \lambda I_t)$ (e.g., [15, Eq. (12)]), which implies $\Phi_t^T (\Phi_t \Phi_t^T + \lambda I_t)^{-1} = (\Phi_t^T \Phi_t + \lambda I)^{-1} \Phi_t^T$. $\square$

Some of the most commonly used kernels are:

- Linear kernel: $k_{\text{lin}}(x, x') = x^T x'$,

- Squared exponential kernel: $k_{\text{SE}}(x, x') = \exp\left(-\frac{\|x - x'\|^2}{2l^2}\right)$,

- Matérn kernel: $k_{\text{Mat}}(x, x') = \frac{2^{1-\nu}}{\Gamma(\nu)} \left( \frac{\sqrt{2\nu}\|x-x'\|}{l} \right) J_\nu \left( \frac{\sqrt{2\nu}\|x-x'\|}{l} \right),$

where $l$ denotes the length-scale hyperparameter, $\nu > 0$ is an additional hyperparameter that dictates the smoothness, and $J(\cdot)$ and $\Gamma(\cdot)$ denote the modified Bessel function and the Gamma function, respectively.

**Maximum information gain.** The maximum information gain is defined as [43]

$$\gamma_t := \max_{A \subseteq \mathcal{X} : |A| = t} I(f_A; y_A)$$

$$= \max_{x_1, \dots, x_t} \frac{1}{2} \log \det(I_t + \lambda^{-1} K_t),$$

where $f_A = [f(x_t)]_{x_t \in A}$, $y_A = [y_t]_{x_t \in A}$, and $I(\cdot; \cdot)$ denotes mutual information. The maximum information gain quantifies the maximum reduction in uncertainty about $f$ after $t$ observations. The following upper bounds for specific kernels have been shown previously [43, 46]:

- Linear kernel: $\gamma_t^{\text{lin}} = O^*(d \log t)$,
- Squared exponential kernel: $\gamma_t^{\text{SE}} = O^*((\log t)^d)$,
- Matérn kernel: $\gamma_t^{\text{Mat}} = O^*\big(t^{\frac{d}{2\nu+d}}\big)$.

The following lemma shows that $\sum_{t=1}^{T} \sigma_{t-1}(x_t)$ can be upper bounded in terms of $\gamma_T$.

**Lemma 9.** *With $\sigma_{t-1}(x_t)$ denoting the posterior standard deviation at $x_t$ based on $(x_1, \dots, x_{t-1})$, we have*

$$\sum_{t=1}^{T} \sigma_{t-1}(x_t) \leq \sqrt{T \sum_{t=1}^{T} \sigma_{t-1}^2(x_t)} \leq \sqrt{\frac{2}{\log(1+\lambda^{-1})} T \gamma_T} \leq \sqrt{(2\lambda+1)T\gamma_T}.$$

*Proof.* The first inequality follows by Cauchy-Schwartz inequality; the second inequality follows from Lemma 5.4 of [43]; the last inequality follows since $(2\lambda + 1)\log(1 + \lambda^{-1}) > 2$ for $\lambda > 0$. $\square$

## B   Corrupted Confidence Bounds

For convenience, we first restate our main assumption regarding non-corrupted confidence bounds.

**Assumption 1** (Regular confidence bounds). *Let $\mu^{(h)}(x)$ and $\sigma^{(h)}(x)$ denote the posterior mean and standard deviation computed (hypothetically) using only the non-corrupted observations $\{(x_i, y_i)\}_{i=1}^{u_h}$ in epoch $h$ using Eqs. (5) and (6). We assume that given $\delta \in (0, 1)$, there exists a sequence of parameters $\beta_h = \beta_h(\delta)$ which is non-decreasing in $h$ and yields with probability at least $1 - \delta$ that*

$$|\mu^{(h)}(x) - f(x)| \leq \beta_h \sigma^{(h)}(x) \tag{9}$$

*simultaneously for all $h \geq 0$ and $x \in \mathcal{X}$.*

In this appendix, we prove Lemma 2, which is restated as follows.

**Lemma 2** (Corrupted confidence bounds). *Under Assumption 1, let $\widetilde{\mu}^{(h)}(x)$ denote the posterior mean based on only the corrupted observations $\{(x_i, \widetilde{y}_i)\}_{i=1}^{u_h}$ in epoch $h$ using Eq. (7), and let $u_{\min} \geq 1$ denote the minimum number of times any single action from $\{x_i\}_{i=1}^{u_h}$ is played, i.e., $u_{\min} = \min_{x \in \{x_1, \dots, x_{u_h}\}} \sum_{i=1}^{u_h} \mathbb{1}\{x_i = x\}$. Then, with probability at least $1 - \delta$, it holds for all $x \in \mathcal{X}$ and $h \geq 0$ that*

$$|\widetilde{\mu}^{(h)}(x) - f(x)| \leq \left( \beta_h + \frac{C\sqrt{u_h}}{u_{\min}\lambda} \right) \sigma^{(h)}(x). \tag{10}$$

*Proof.* For simplicity, we denote the epoch length $u_h$ by $t$ in this proof, and use $\mu_t(\cdot), \widetilde{\mu}_t(\cdot),$ and $\sigma_t(\cdot)$ to denote $\mu^{(h)}(\cdot), \widetilde{\mu}^{(h)}(\cdot),$ and $\sigma^{(h)}(\cdot)$, respectively. Thus, here $\sigma_t(\cdot)$ is defined with respect to the $t = u_h$ sampled points, whereas Algorithm 1 only computes the posterior variance with respect to the points selected in the for loop, of which there are $l_h$ (possibly strictly fewer than $u_h$). This part of the analysis only requires the former notion, so there should be no confusion between the two.

We first recall the definition of the robust-corrupted mean estimator from Eq. (7), i.e.,

$$\widetilde{\mu}_t(x) = k_t(x)^T (K_t + \lambda I_t)^{-1} \widetilde{Y}_t, \tag{24}$$

where $\widetilde{Y}_t \in \mathbb{R}^t$ and $\widetilde{Y}_t[i] = \frac{\sum_{j=1}^t \mathbb{1}\{x_i = x_j\} \widetilde{y}_j}{\sum_{j=1}^t \mathbb{1}\{x_i = x_j\}}$ for $i \in [t]$. We use $z_t(x; \lambda) \in \mathbb{R}^t$ to denote $k_t(x)^T (K_t + \lambda I_t)^{-1}$ which implies $\widetilde{\mu}_t(x) = \sum_{i=1}^t z_t(x; \lambda)[i] \cdot \widetilde{Y}_t[i]$.

We will also use the following equivalent feature-based expression: $z_t(x; \lambda) = k_t(x)^T (K_t + \lambda I)^{-1} = \phi(x)^T \Phi_t^T (\Phi_t \Phi_t^T + \lambda I_t)^{-1}$, where $k(x, x') = \phi(x)^T \phi(x')$, $\phi(x) \in \mathcal{H}_k(\mathcal{X})$ for every $x \in \mathcal{X}$, and $\Phi_t = (\phi(x_{t'}))_{t' \leq t}$ denotes the matrix of (potentially infinite-dimensional) features placed in $t$ rows. Finally, recalling that $I$ denotes the infinite-dimensional identity matrix in feature space, we also have

$$z_t(x; \lambda) = \phi(x)^T (\Phi_t^T \Phi_t + \lambda I)^{-1} \Phi_t^T, \tag{25}$$

which follows from the standard push-through identity $\Phi_t^T \left( \Phi_t \Phi_t^T + \lambda I_t \right)^{-1} = \left( \Phi_t^T \Phi_t + \lambda I \right)^{-1} \Phi_t^T$ (e.g., see Eq. (12) of [15]).

We proceed to analyze the corrupted estimator $\widetilde{\mu}_t(x)$:

$$\widetilde{\mu}_t(x) = \sum_{i=1}^t z_t(x; \lambda)[i] \, \widetilde{Y}_t[i] \tag{26}$$

$$= \sum_{i=1}^t \frac{\sum_{j=1}^t \mathbb{1}\{x_i = x_j\} \widetilde{y}_j}{\sum_{j=1}^t \mathbb{1}\{x_i = x_j\}} z_t(x; \lambda)[i] \tag{27}$$

$$= \sum_{i=1}^t \frac{\sum_{j=1}^t \mathbb{1}\{x_i = x_j\}(f(x_i) + \epsilon_j + c_j)}{\sum_{j=1}^t \mathbb{1}\{x_i = x_j\}} z_t(x; \lambda)[i] \tag{28}$$

$$= \sum_{i=1}^t f(x_i) z_t(x; \lambda)[i] + \sum_{i=1}^t \frac{\sum_{j=1}^t \mathbb{1}\{x_i = x_j\} \epsilon_j}{\sum_{j=1}^t \mathbb{1}\{x_i = x_j\}} z_t(x; \lambda)[i] + \sum_{i=1}^t \frac{\sum_{j=1}^t \mathbb{1}\{x_i = x_j\} c_j}{\sum_{j=1}^t \mathbb{1}\{x_i = x_j\}} z_t(x; \lambda)[i] \tag{29}$$

$$= \sum_{i=1}^t f(x_i) z_t(x; \lambda)[i] + \sum_{i=1}^t \epsilon_i z_t(x; \lambda)[i] + \sum_{i=1}^t \frac{\sum_{j=1}^t \mathbb{1}\{x_i = x_j\} c_j}{\sum_{j=1}^t \mathbb{1}\{x_i = x_j\}} z_t(x; \lambda)[i] \tag{30}$$

$$= \sum_{i=1}^t (f(x_i) + \epsilon_i) z_t(x; \lambda)[i] + \sum_{i=1}^t \frac{\sum_{j=1}^t \mathbb{1}\{x_i = x_j\} c_j}{\sum_{j=1}^t \mathbb{1}\{x_i = x_j\}} z_t(x; \lambda)[i] \tag{31}$$

$$= \mu_t(x) + \sum_{i=1}^t \frac{\sum_{j=1}^t \mathbb{1}\{x_i = x_j\} c_j}{\sum_{j=1}^t \mathbb{1}\{x_i = x_j\}} z_t(x; \lambda)[i]. \tag{32}$$

Here, we used the definition of $\widetilde{Y}_t[i]$ in Eq. (27) and the corrupted observation $\widetilde{y}_j$ corresponding to $x_j = x_i$ at time $j$ in Eq. (28), while Eq. (29) follows from rearranging. The proof of Eq. (30) is deferred to the next paragraph. Finally, Eq. (32) follows from the definition of the noisy stochastic observation $y_i = f(x_i) + \epsilon_i$ and the definition of the standard (non-corrupted) mean estimator from Eq. (5).

To prove Eq. (30), we define $\widetilde{\epsilon}_t \in \mathbb{R}^t$ such that $\widetilde{\epsilon}_t[i] = \frac{\sum_{j=1}^t \mathbb{1}\{x_i=x_j\}\epsilon_j}{\sum_{j=1}^t \mathbb{1}\{x_i=x_j\}}$ for $i \in [t]$, and use $u_t(x)$ to denote $\sum_{j=1}^t \mathbb{1}\{x = x_j\}$, i.e., the number of times action $x$ was played during the $t$ rounds. Then,

$$\sum_{i=1}^t \frac{\sum_{j=1}^t \mathbb{1}\{x_i=x_j\}\epsilon_j}{\sum_{j=1}^t \mathbb{1}\{x_i=x_j\}} z_t(x; \lambda)[i] = z_t(x; \lambda)\widetilde{\epsilon}_t \tag{33}$$

$$= \phi(x)^T (\Phi_t^T \Phi_t + \lambda I)^{-1} \Phi_t^T \widetilde{\epsilon}_t \tag{34}$$

$$= \phi(x)^T (\Phi_t^T \Phi_t + \lambda I)^{-1} \sum_{i=1}^t \phi(x_i)\widetilde{\epsilon}_t[i] \tag{35}$$

$$= \phi(x)^T (\Phi_t^T \Phi_t + \lambda I)^{-1} \sum_{x \in \mathcal{X}, u_t(x) \neq 0} u_t(x)\phi(x)\frac{\sum_{j=1}^t \mathbb{1}\{x=x_j\}\epsilon_j}{u_t(x)} \tag{36}$$

$$= \phi(x)^T (\Phi_t^T \Phi_t + \lambda I)^{-1} \sum_{x \in \mathcal{X}, u_t(x) \neq 0} \phi(x) \sum_{j=1}^t \mathbb{1}\{x = x_j\}\epsilon_j \tag{37}$$

$$= \phi(x)^T (\Phi_t^T \Phi_t + \lambda I)^{-1} \sum_{j=1}^t \phi(x_j)\epsilon_j \tag{38}$$

$$= \phi(x)^T (\Phi_t^T \Phi_t + \lambda I)^{-1} \Phi_t^T \epsilon_t \tag{39}$$

$$= z_t(x; \lambda)\epsilon_t = \sum_{i=1}^t \epsilon_i z_t(x; \lambda)[i], \tag{40}$$

where Eq. (34) holds due to Eq. (25), and Eq. (36) uses the definitions of $\widetilde{\epsilon}_t$ and $u_t(x)$, and (38)–(40) are analogous to (33)–(35) in the opposite order.

By rearranging Eq. (32), it follows that we can bound the absolute difference between the corrupted mean estimator and the standard one as follows:

$$|\widetilde{\mu}_t(x) - \mu_t(x)| \leq \left| \sum_{i=1}^t \frac{\sum_{j=1}^t \mathbb{1}\{x_i=x_j\}c_j}{\sum_{j=1}^t \mathbb{1}\{x_i=x_j\}} z_t(x; \lambda)[i] \right|. \tag{41}$$

Next, we proceed to analyze the right hand side term. We use $C_t$ to denote a vector in $\mathbb{R}^t$ such that $C_t[i] = \frac{\sum_{j=1}^t \mathbb{1}\{x_i=x_j\}c_j}{\sum_{j=1}^t \mathbb{1}\{x_i=x_j\}}$ for every $i \in [t]$. Then, continuing from Eq. (41), we have

$$\left| \sum_{i=1}^t \frac{\sum_{j=1}^t \mathbb{1}\{x_i=x_j\}c_j}{\sum_{j=1}^t \mathbb{1}\{x_i=x_j\}} z_t(x; \lambda)[i] \right| = \left| \phi(x)^T \underbrace{(\Phi_t^T \Phi_t + \lambda I)^{-1}}_{:=\Gamma_t^{-1}} \Phi_t^T C_t \right| \tag{42}$$

$$= \left| \sum_{i=1}^t C_t[i]\phi(x)^T \Gamma_t^{-1}\phi(x_i) \right|, \tag{43}$$

where we again used the form of $z_t$ given in Eq. (25).

Let $C_t(x) = \sum_{j=1}^{t} \mathbb{1}\{x = x_j\}c_j$ for $x \in \mathcal{X}$. Then, we can rewrite (43) as

$$\Big|\sum_{i=1}^{t}\frac{\sum_{j=1}^{t}\mathbb{1}\{x_i=x_j\}c_j}{\sum_{j=1}^{t}\mathbb{1}\{x_i=x_j\}}z_t(x;\lambda)[i]\Big| = \Big|\sum_{x'\in\mathcal{X},u_t(x')\neq 0}\frac{C_t(x')}{u_t(x')}u_t(x')\phi(x)^T\Gamma_t^{-1}\phi(x')\Big| \tag{44}$$

$$\leq \sum_{x'\in\mathcal{X},u_t(x')\neq 0}\frac{C}{u_t(x')}u_t(x')\big|\phi(x)^T\Gamma_t^{-1}\phi(x')\big| \tag{45}$$

$$\leq \frac{C}{u_{\min}}\sum_{x'\in\mathcal{X},u_t(x')\neq 0}u_t(x')\big|\phi(x)^T\Gamma_t^{-1}\phi(x')\big| \tag{46}$$

$$\leq \frac{C}{u_{\min}}\sqrt{\Big(\sum_{x'\in\mathcal{X},u_t(x')\neq 0}u_t(x')\Big)\phi(x)^T\sum_{x'\in\mathcal{X},u_t(x')\neq 0}u_t(x')\Gamma_t^{-1}\phi(x')\phi(x')^T\Gamma_t^{-1}\phi(x)} \tag{47}$$

$$\leq \frac{C}{u_{\min}}\sqrt{\Big(\sum_{x'\in\mathcal{X},u_t(x')\neq 0}u_t(x')\Big)\phi(x)^T\sum_{x'\in\mathcal{X},u_t(x')\neq 0}u_t(x')\Gamma_t^{-1}\big(\phi(x')\phi(x')^T + \tfrac{\lambda}{t}I\big)\Gamma_t^{-1}\phi(x)} \tag{48}$$

$$= \frac{C}{u_{\min}}\sqrt{\sum_{x'\in\mathcal{X},u_t(x')\neq 0}u_t(x')\|\phi(x)\|^2_{\Gamma_t^{-1}}} \tag{49}$$

$$= \frac{C}{u_{\min}}\sqrt{t}\|\phi(x)\|_{\Gamma_t^{-1}} = \frac{C\sqrt{t}}{\lambda u_{\min}}\sigma_t(x), \tag{50}$$

where:

- Eq. (45) holds since $C \geq |C_t(x)|$ for every $x \in \mathcal{X}$.

- Eq. (46) follows from the definition of $u_{\min}$ in the lemma statement.

- To obtain Eq. (47), we multiply and divide by $\sum_{x\in\mathcal{X},u_t(x)\neq 0}u_t(x)$ and apply $\mathbb{E}[|X|] \leq \sqrt{\mathbb{E}[X^2]}$ considering the distribution $\frac{u_t(x')}{\sum_{x\in\mathcal{X},u_t(x)\neq 0}u_t(x)}$. (Note also that, in generic vector-matrix notation, $(a^TMb)^2 = a^TMbb^TMa$ when $M$ is a symmetric matrix. )

- To obtain Eq. (49), we use $\sum_{x'\in\mathcal{X},u_t(x')\neq 0}u_t(x')\frac{\lambda}{t}I = \lambda I$ (i.e., $\sum_{x'\in\mathcal{X},u_t(x')\neq 0}u_t(x') = t$), and note that $\Gamma_t = \big(\sum_{x'\in\mathcal{X},u_t(x')\neq 0}u_t(x')\phi(x')\phi(x')^T\big) + \lambda I$. Combining these facts gives $\sum_{x'\in\mathcal{X},u_t(x')\neq 0}u_t(x')\big(\phi(x')\phi(x')^T + \frac{\lambda}{t}I\big) = \Gamma_t$, which cancels with one of the $\Gamma_t^{-1}$ terms. The remaining quantity $\phi(x)^T\Gamma_t^{-1}\phi(x)$ is precisely the definition of $\|\phi(x)\|^2_{\Gamma_t^{-1}}$.

- Finally, Eq. (50) holds since
$$\|\phi(x)\|^2_{\Gamma_t^{-1}} = \phi(x)^T\Gamma_t^{-1}\phi(x) = \lambda^{-1}\sigma_t^2(x), \tag{51}$$
which holds due to Eq. (16).

Conditioned on the event in Assumption 1, the final result then follows since

$$|\widetilde{\mu}_t(x) - f(x)| \leq |\mu_t(x) - f(x)| + |\widetilde{\mu}_t(x) - \mu_t(x)| \leq \Big(\beta_h + \frac{C\sqrt{t}}{\lambda u_{\min}}\Big)\sigma_t(x), \tag{52}$$

where we apply Assumption 1 and Eq. (50) to upper bound $|\mu_t(x) - f(x)|$ and $|\widetilde{\mu}_t(x) - \mu_t(x)|$, respectively.

$\square$

# C  Auxiliary Results

In the following, we recall the notation in Algorithm 1, particularly the truncation parameter $\psi > 0$. In addition, in accordance with the algorithm statement, quantities such as $\sigma_t(\cdot)$ and $K_t$ implicitly

depend on $h$, and are defined with respect to the $t \leq l_h$ points chosen up to time $t$ in the for loop (as opposed to the $u_h \geq l_h$ points sampled *after* the for loop).

We first formalize the claim that the number of epochs is at most $\bar{H} = \log_2 T$.

**Lemma 10.** *For any time horizon $T$, Algorithm 1 terminates after at most $\log_2 T$ epochs.*

*Proof.* This follows immediately from the fact that we initialize $l_0 = 2$, double $l_h$ after each epoch, and take at least $l_h$ actions in epoch $h$ (see Line 12 with $\sum_x \xi_h(x) = 1$) until $T$ actions have been played. $\qquad\square$

Next, we state a simple result regarding the epoch lengths.

**Lemma 11.** *The length $u_h$ of epoch $h$ in Algorithm 1 satisfies $u_h \leq l_h(2 + |\mathcal{S}_h|\psi)$.*

*Proof.* The number of times each action from $\mathcal{S}_h$ is played is $u_h(x)$, and is given in Algorithm 1 (Line 12). Hence, we have

$$u_h = \sum_{x \in \mathcal{S}_h} \lceil l_h \max\{\xi_h(x), \psi\} \rceil \tag{53}$$

$$\leq \sum_{x \in \mathcal{S}_h} (l_h \max\{\xi_h(x), \psi\} + 1) \tag{54}$$

$$\leq |\mathcal{S}_h| + \sum_{x \in \mathcal{S}_h} (l_h \xi_h(x) + l_h \psi) \tag{55}$$

$$\leq 2l_h + l_h \psi |\mathcal{S}_h| = l_h(2 + \psi|\mathcal{S}_h|), \tag{56}$$

where in the last inequality, we use $|\mathcal{S}_h| \leq l_h$ and $\sum_{x \in \mathcal{S}_h} \xi_h(x) = 1$. $\qquad\square$

The following result characterizes the posterior uncertainty of points sampled in between the switching events in Algorithm 1, and may be of independent interest for problems in RKHS function spaces, particularly in settings where infrequent action switching is desirable.

**Lemma 12.** *Consider any epoch $h$, the corresponding set of actions $\mathcal{X}_h$, and the regularization parameter $\lambda > 0$. Let $t, t' \in [l_h]$ denote two rounds in epoch $h$ such that $t \geq t'$, and for which*

$$\det(I_t + \lambda^{-1}K_t) \leq \eta \det(I_{t'} + \lambda^{-1}K_{t'}) \tag{57}$$

*(i.e., the condition in Line 6 in Algorithm 1 does not hold), where $\eta > 1$. Then, for every $x \in \mathcal{X}_h$, it holds that*

$$\sigma_{t'}(x) \leq \sqrt{\eta}\sigma_t(x). \tag{58}$$

*Proof.* We first consider the case that $k(x, x') = \phi(x)^T \phi(x')$ for every $x, x' \in \mathcal{X}$ with finite-dimensional features: $\phi(x) \in \mathbb{R}^{d_\phi}$ for some $d_\phi < \infty$. We let $\Phi_t = (\phi(x_{t'}))_{t' \leq t} \in \mathbb{R}^{t \times d_\phi}$ denote the matrix of features placed in $t$ rows. We will later drop the assumption of finite dimensionality to obtain the result in our original setup.

We also note that if $\phi(x)$ contains all zeros for some input $x \in \mathcal{X}$, the statement in Equation (58) trivially holds (i.e., both sides are zero), so in the rest of the analysis, we assume that this is not the case.

In the following, let $x$ be any fixed point in the domain. From Eq. (57), we have:

$$\eta \geq \frac{\det(\lambda^{-1}K_t + I_t)}{\det(\lambda^{-1}K_{t'} + I_{t'})} \tag{59}$$

$$= \frac{\det(K_t + \lambda I_t)}{\det(K_{t'} + \lambda I_{t'})} \tag{60}$$

$$= \frac{\det\left(\Phi_t^T \Phi_t + \lambda I_d\right)}{\det\left(\Phi_{t'}^T \Phi_{t'} + \lambda I_d\right)} \tag{61}$$

$$= \frac{\det\left(\left(\Phi_{t'}^T \Phi_{t'} + \lambda I_d\right)^{-1}\right)}{\det\left(\left(\Phi_t^T \Phi_t + \lambda I_d\right)^{-1}\right)} \tag{62}$$

$$\geq \frac{\phi(x)^T \left(\Phi_{t'}^T \Phi_{t'} + \lambda I_d\right)^{-1} \phi(x)}{\phi(x)^T \left(\Phi_t^T \Phi_t + \lambda I_d\right)^{-1} \phi(x)} \tag{63}$$

$$= \frac{\sigma_{t'}^2(x)}{\sigma_t^2(x)}. \tag{64}$$

Here, Eq. (61) holds due to the Weinstein–Aronszajn identity (i.e., $\det(I + AB) = \det(I + BA)$), and in Eq. (62) we use the fact that $\det(A) = (\det(A^{-1}))^{-1}$ for any invertible matrix $A$. Eq. (63) is proved in the following paragraph, and Eq. (64) follows from the alternative definition of $\sigma_t(\cdot)$ in Eq. (16).

It remains to prove the inequality in Eq. (63), which closely follows the proof of [2, Lemma 12]. For any $i \in [t]$, let $V_i := \lambda^{-1} \Phi_i^T \Phi_i + I$. We first show that

$$\frac{\phi(x)^T V_t \phi(x)}{\phi(x)^T V_{t-1} \phi(x)} \leq 1 + \|\lambda^{-1/2} \phi(x_t)\|_{V_{t-1}^{-1}}^2. \tag{65}$$

We have for any $x \in \mathcal{X}_h$ that

$$\phi(x)^T V_t \phi(x) = \phi(x)^T V_{t-1} \phi(x) + \phi(x)^T \left(\lambda^{-1} \phi(x_t) \phi(x_t)^T\right) \phi(x) \tag{66}$$

$$= \phi(x)^T V_{t-1} \phi(x) + \lambda^{-1} \left(\phi(x)^T \phi(x_t)\right)^2 \tag{67}$$

$$= \phi(x)^T V_{t-1} \phi(x) + \lambda^{-1} \left(\phi(x)^T V_{t-1}^{1/2} V_{t-1}^{-1/2} \phi(x_t)\right)^2 \tag{68}$$

$$\leq \phi(x)^T V_{t-1} \phi(x) + \lambda^{-1} \|\phi(x)^T V_{t-1}^{1/2}\|_2^2 \|V_{t-1}^{-1/2} \phi(x_t)\|_2^2 \tag{69}$$

$$= \phi(x)^T V_{t-1} \phi(x) + \lambda^{-1} (\phi(x)^T V_{t-1} \phi(x))(\phi(x_t) V_{t-1}^{-1} \phi(x_t)) \tag{70}$$

$$= \left(1 + \|\lambda^{-1/2} \phi(x_t)\|_{V_{t-1}^{-1}}^2\right) \phi(x)^T V_{t-1} \phi(x), \tag{71}$$

where Eq. (69) follows from Cauchy-Schwarz inequality. Hence, Eq. (65) follows by rearranging.

Since $t > t'$, we have:

$$\frac{\phi(x)^T V_t \phi(x)}{\phi(x)^T V_{t'} \phi(x)}$$

$$= \frac{\phi(x)^T V_t \phi(x)}{\phi(x)^T V_{t-1} \phi(x)} \cdot \frac{\phi(x)^T V_{t-1} \phi(x)}{\phi(x)^T V_{t-2} \phi(x)} \cdot \ldots \cdot \frac{\phi(x)^T V_{t'+1} \phi(x)}{\phi(x)^T V_{t'} \phi(x)} \tag{72}$$

$$\leq \left(1 + \|\lambda^{-1/2} \phi(x_t)\|_{V_{t-1}^{-1}}^2\right) \cdot \left(1 + \|\lambda^{-1/2} \phi(x_{t-1})\|_{V_{t-2}^{-1}}^2\right) \cdot \ldots \cdot \left(1 + \|\lambda^{-1/2} \phi(x_{t'+1})\|_{V_{t'}^{-1}}^2\right) \tag{73}$$

$$= \frac{\det(V_t)}{\det(V_{t-1})} \cdot \frac{\det(V_{t-1})}{\det(V_{t-2})} \cdot \ldots \cdot \frac{\det(V_{t'+1})}{\det(V_{t'})} \tag{74}$$

$$= \frac{\det(V_t)}{\det(V_{t'})}, \tag{75}$$

where Eq. (73) follows from Eq. (65), and Eq. (74) uses the fact that

$$\frac{\det(V_t)}{\det(V_{t-1})} = 1 + \|\lambda^{-1/2} \phi(x_t)\|_{V_{t-1}^{-1}}^2, \tag{76}$$

which is shown in the proof of [18, Theorem 2.2].

It remains to handle the possibly infinite feature dimension. Consider $k(x, x') = \sum_{i=1}^{\infty} \lambda_i \phi_i(x) \phi_i(x')$ and let $k_{d_\phi}(x, x') = \sum_{i=1}^{d_\phi} \lambda_i \phi_i(x) \phi_i(x')$ denote the finite dimensional kernel that corresponds to the $d_\phi$-dimensional feature space such that $\lim_{d_\phi \to \infty} k_{d_\phi}(x, x') = k(x, x')$ for every $x, x' \in \mathcal{X}$. We use $K_{t,d_\phi}$ and $\sigma^2_{t,d_\phi}(\cdot)$ to denote the restriction of the corresponding quantities when the kernel $k_{d_\phi}(\cdot, \cdot)$ is used. First, we note that Eq. (60) still holds. Moreover, we have $\frac{\det(K_t + \lambda I_t)}{\det(K_{t'} + \lambda I_{t'})} = \lim_{d_\phi \to \infty} \frac{\det(K_{t,d_\phi} + \lambda I_t)}{\det(K_{t',d_\phi} + \lambda I_{t'})}$ and $\frac{\sigma^2_{t'}(x)}{\sigma^2_t(x)} = \lim_{d_\phi \to \infty} \frac{\sigma^2_{t',d_\phi}(x)}{\sigma^2_{t,d_\phi}(x)}$, and the former limit is lower bounded by the latter due to the fact that Eqs. (61) to (63) are all valid for the finite $d_\phi$-feature approximation. Thus, the final result still holds for infinite dimensional kernels. $\qquad \square$

Next, we uniformly bound the posterior variance for the points remaining after a given epoch.

**Lemma 13.** *For any epoch $h$ and the corresponding set of actions $\mathcal{X}_h$, it holds that*

$$\max_{x \in \mathcal{X}_h} \sigma^{(h)}(x) \leq \sqrt{\frac{\eta(2\lambda + 1)\gamma_{l_h}}{l_h}}. \tag{77}$$

*Proof.* Recall that $u_h$ corresponds to the length of epoch $h$ and that we can $\sigma^{(h)}(x)$ represents a posterior variance $\sigma_{u_h}(x)$ taken with respect to the $u_h$ sampled points after the epoch. We first relate this to the posterior variance $\sigma_{l_h}(x)$ (abusing notation slightly) taken only with respect to the $l_h$ points in the for loop in Algorithm 1. In particular, we claim that the former is upper bounded by the latter, and so it suffices to work with $\sigma_{l_h}(x)$. To see this, we recall that each $x$ is sampled $u_h(x) = \lceil l_h \max\{\xi_h(x), \psi\} \rceil$ times, and the definition $\xi_h(x) = \frac{\sum_{i=1}^{l_h} \mathbb{1}\{x = x_i\}}{l_h}$ gives $l_h \xi_h(x) = \sum_{i=1}^{l_h} \mathbb{1}\{x = x_i\}$. Thus, the number of times each point is sampled is at least as high as the number of times it is selected in the for loop. Since conditioning on a higher number of points always decreases (or at least does not increase) the posterior variance in a Gaussian process, the desired claim follows.

We proceed to upper bound $\max_{x \in \mathcal{X}_h} \sigma_{l_h}(x)$. Let $\mathcal{T}_h = \{t \in [l_h] : \det(I_t + \lambda^{-1} K_t) > \eta \det(I_{t'} + \lambda^{-1} K_{t'})\}$ be the rounds in which the condition in Line 6 (Algorithm 1) is satisfied. Moreover, let $\bar{\mathcal{T}}_h = \mathcal{T}_h \cup \{0\}$ and let its elements $\bar{\mathcal{T}}_h = \{t'_0, \ldots, t'_i, \ldots, t'_{|\mathcal{T}_h|}\}$ be increasingly ordered. We note that $\max_{x \in \mathcal{X}_h} \sigma_{l_h}(x) \leq \sigma_{t'_i}(x_{t'_i + 1})$ for every $t'_i \in \bar{\mathcal{T}}_h$ according to the selection rule in Algorithm 1 (Line 4) and the fact that $\sigma_t(\cdot)$ is decreasing with respect to $t$. It follows that

$$l_h \left( \max_{x \in \mathcal{X}_h} \sigma_{l_h}(x) \right) \leq \left( \sum_{i=0}^{|\mathcal{T}_h| - 1} (t'_{i+1} - t'_i) \sigma_{t'_i}(x_{t'_i + 1}) \right) + (l_h - t'_{|\mathcal{T}_h|}) \sigma_{t'_{|\mathcal{T}_h|}}(x_{t'_{|\mathcal{T}_h|} + 1}). \tag{78}$$

Observe that by definition, we have $x_{t'_i + 1} = x_{t'_i + 2} = \cdots = x_{t'_{i+1}}$, i.e., these form a chain of identical points up to when the switching condition in Line 6 holds. Accordingly, by Lemma 12, it holds that $\sigma_{t'_i}(x_{t'_i + 1}) \leq \sqrt{\eta} \sigma_t(x_{t+1})$ for every $t \in \{t'_i, \ldots, t'_{i+1} - 1\}$. By combining this with Eq. (78), we obtain

$$l_h \left( \max_{x \in \mathcal{X}_h} \sigma_{l_h}(x) \right) \leq \sqrt{\eta} \sum_{t=0}^{l_h - 1} \sigma_t(x_{t+1}). \tag{79}$$

Finally, from Lemma 9, we have $\sum_{t=0}^{l_h - 1} \sigma_t(x_{t+1}) \leq \sqrt{(2\lambda + 1)\gamma_{l_h} l_h}$. By combining this with Equation (79) and rearranging, we obtain the final result. $\qquad \square$

Finally, we provide a result bounding the size of the set $\mathcal{S}_h$ in Algorithm 1.

**Lemma 14.** *For any epoch $h$ and the corresponding set $\mathcal{S}_h$, we have*

$$|\mathcal{S}_h| \leq \frac{2}{\ln \eta} \gamma_T. \tag{80}$$

*Proof.* By the algorithm design, the set $\mathcal{S}_h$ grows by at most one element after the condition in Line 6 is satisfied, i.e., when

$$\det(I_t + \lambda^{-1}K_t) > \eta \det(I_{t'} + \lambda^{-1}K_{t'}), \tag{81}$$

where $t$ is the current iteration, and $t'$ is iteration prior to $t$ for which Line 6 held (or $t' = 0$). As before, let $\mathcal{T}_h = \{t \in [l_h] : \det(I_t + \lambda^{-1}K_t) > \eta \det(I_{t'} + \lambda^{-1}K_{t'})\}$ be the rounds in which this holds, ordered with respect to time. Thus, for consecutive $t_i$ and $t_{i-1}$ belonging to $\mathcal{T}_h$, we have

$$\det(I_{t_i} + \lambda^{-1}K_{t_i}) > \eta \det(I_{t_{i-1}} + \lambda^{-1}K_{t_{i-1}}). \tag{82}$$

By applying the previous relation recursively, it follows that

$$\begin{aligned}
\det(I_{t_i} + \lambda^{-1}K_{t_i}) &> \eta \det(I_{t_{i-1}} + \lambda^{-1}K_{t_{i-1}}) \\
&> \eta^2 \det(I_{t_{i-2}} + \lambda^{-1}K_{t_{i-2}}) \\
&> \ldots \\
&> \eta^{i+1} \det(1 + \lambda^{-1}) = \eta^{i+1}(1 + \lambda^{-1}).
\end{aligned} \tag{83}$$

Using the definition of $\gamma_{l_h}$ given in (8), and noting that the size of the set $\mathcal{T}_h$ is at least $|\mathcal{S}_h| - 1$, we obtain

$$\gamma_{l_h} \geq \tfrac{1}{2} \ln \det(I_{l_h} + \lambda^{-1}K_{l_h}) \geq \tfrac{1}{2} \ln(\eta^{|\mathcal{S}_h|}(1 + \lambda^{-1})) \geq \tfrac{1}{2} \ln(\eta^{|\mathcal{S}_h|}). \tag{84}$$

By rearranging, we obtain

$$|\mathcal{S}_h| \leq \tfrac{2}{\ln \eta} \gamma_{l_h}. \tag{85}$$

The result then follows since $\gamma_T \geq \gamma_{l_h}$ for every $h$.

$\square$

# D   Regret Analysis

In this appendix, we prove our main result, Theorem 3. We first upper bound the regret of any point sampled in a given epoch.

**Lemma 15.** *With probability at least $1 - \delta$, we have for every epoch $h$ and $x \in \mathcal{X}_h$ that*

$$\max_{x \in \mathcal{X}_h} f(x) - f(x) \leq 4\Big(\beta_{h-1} + \tfrac{C\sqrt{u_{h-1}}}{l_{h-1}\psi\lambda}\Big)\sqrt{\frac{\eta(2\lambda+1)\gamma_{l_{h-1}}}{l_{h-1}}}. \tag{86}$$

*Proof.* Recall that $u_h$ denotes the epoch length, and let $x_h^* \in \arg\max_{x \in \mathcal{X}_h} f(x)$. By using the validity of the confidence bounds from the end of the previous epoch $h - 1$ (see Lemma 2), we have for all $x \in \mathcal{X}_h$ that

$$\begin{aligned}
f(x_h^*) - f(x) \leq {}& \widetilde{\mu}^{(h-1)}(x_h^*) + \big(\beta_{h-1} + \tfrac{C}{l_{h-1}\psi\lambda}\sqrt{u_{h-1}}\big)\sigma^{(h-1)}(x^*) \\
& - \widetilde{\mu}^{(h-1)}(x) + \big(\beta_{h-1} + \tfrac{C}{l_{h-1}\psi\lambda}\sqrt{u_{h-1}}\big)\sigma^{(h-1)}(x), \quad (87)
\end{aligned}$$

where in Lemma 2 we substitute $h - 1$ and set $u_{\min} = l_{h-1}\psi$ (since each action selected in epoch $h - 1$ in Algorithm 1 is played at least $\lceil l_{h-1}\psi \rceil$ times), to upper and lower bound $\max_{x \in \mathcal{X}_h} f(x)$ and $f(x)$, respectively.

Next, for any $x \in \mathcal{X}_h$, it holds that

$$\begin{aligned}
& \widetilde{\mu}^{(h-1)}(x) + \big(\beta_{h-1} + \tfrac{C}{l_{h-1}\psi\lambda}\sqrt{u_{h-1}}\big)\sigma^{(h-1)}(x) \\
& \geq \max_{x \in \mathcal{X}_{h-1}} \Big(\widetilde{\mu}^{(h-1)}(x) - \big(\beta_{h-1} + \tfrac{C}{l_{h-1}\psi\lambda}\sqrt{u_{h-1}}\big)\sigma^{(h-1)}(x)\Big) \quad (88) \\
& \geq \widetilde{\mu}^{(h-1)}(x_h^*) - \big(\beta_{h-1} + \tfrac{C}{l_{h-1}\psi\lambda}\sqrt{u_{h-1}}\big)\sigma^{(h-1)}(x_h^*), \quad (89)
\end{aligned}$$

where Eq. (88) follows from the elimination condition (see Line 15 in Algorithm 1), and Eq. (89) holds since $x_h^* \in \mathcal{X}_h \subseteq \mathcal{X}_{h-1}$.

Combining Eq. (89) with Eq. (87), we obtain

$$f(x_h^*) - f(x) \le 2\big(\beta_{h-1} + \tfrac{C}{l_{h-1}\psi\lambda}\sqrt{u_{h-1}}\big)\sigma^{(h-1)}(x_h^*) + 2\big(\beta_{h-1} + \tfrac{C}{l_{h-1}\psi\lambda}\sqrt{u_{h-1}}\big)\sigma^{(h-1)}(x).$$
(90)

$$\le 4\big(\beta_{h-1} + \tfrac{C}{l_{h-1}\psi\lambda}\sqrt{u_{h-1}}\big)\max_{x\in\mathcal{X}_{h-1}}\sigma^{(h-1)}(x).$$
(91)

The desired result then follows by upper bounding $\max_{x\in\mathcal{X}_{h-1}}\sigma^{(h-1)}(x)$ according to Lemma 13. $\square$

We are ready to prove our main theorem, which is restated as follows.

**Theorem 3** (Main result). *Under the preceding setup and Assumption 1, for any corruption budget $C \ge 0$, Algorithm 1 with a constant switching parameter $\eta > 1$ and truncation parameter $\psi = \frac{\ln\eta}{2\gamma_T}$ satisfies the following with probability at least $1 - \delta$:*

$$R_T = O^*\big(\beta_{\bar{H}}\sqrt{T\gamma_T} + C\gamma_T^{3/2}\big).$$
(12)

*Proof.* Throughout the proof, we condition on the confidence bounds from Lemma 2 holding true. We use $u_h(x)$ to denote the number of times action $x$ is played in epoch $h$, and bound the cumulative regret of Algorithm 1 as follows:

$$R_T = \sum_{h=0}^{H-1}\sum_{x\in\mathcal{S}_h}\big(f(x^*) - f(x)\big)u_h(x)$$
(92)

$$\le u_0 B + \sum_{h=1}^{H-1}\sum_{x\in\mathcal{S}_h}\big(f(x^*) - f(x)\big)u_h(x)$$
(93)

$$\le u_0 B + \sum_{h=1}^{H-1}\sum_{x\in\mathcal{S}_h}u_h(x)\cdot 4\big(\beta_{h-1} + \tfrac{C\sqrt{u_{h-1}}}{l_{h-1}\psi\lambda}\big)\sqrt{\tfrac{\eta(2\lambda+1)\gamma_{l_{h-1}}}{l_{h-1}}}.$$
(94)

Here, Eq. (92) follows since only points from $\mathcal{S}_h$ are queried by the algorithm (and each point $x \in \mathcal{S}_h$ is queried $u_h(x)$ times), Eq. (93) follows since the bound on the RKHS norm implies the same bound on the maximal function value when the kernel $k(\cdot,\cdot)$ is normalized (namely, $k(x,x) \le 1$ for every $x$):

$$|f(x)| = |\langle f, k(x,\cdot)\rangle_k| \le \|f\|_k\|k(x,\cdot)\|_k = \|f\|_k\langle k(x,\cdot), k(x,\cdot)\rangle_k^{1/2} \le B\cdot k(x,x)^{1/2} \le B, \quad (95)$$

and Eq. (94) follows from Lemma 15 and by noting that $f(x^*) = \max_{x\in\mathcal{X}_h} f(x)$ for every $h = 0, 1, \ldots, H-1$ (i.e., since the confidence bounds of Lemma 2 are valid, the global maximizer never gets eliminated). Next, from Eq. (94), by noting that $\sum_{x\in\mathcal{S}_h}u_h(x) = u_h$, we have:

$$R_T \le u_0 B + \sum_{h=1}^{H-1}4u_h\big(\beta_{h-1} + \tfrac{C\sqrt{u_{h-1}}}{l_{h-1}\psi\lambda}\big)\sqrt{\tfrac{\eta(2\lambda+1)\gamma_{l_{h-1}}}{l_{h-1}}}$$
(96)

$$\le u_0 B + \sum_{h=1}^{H-1}4l_h(2+\psi|\mathcal{S}_h|)\big(\beta_{h-1} + \tfrac{C\sqrt{l_{h-1}(2+\psi|\mathcal{S}_{h-1}|)}}{l_{h-1}\psi\lambda}\big)\sqrt{\tfrac{\eta(2\lambda+1)\gamma_{l_{h-1}}}{l_{h-1}}}$$
(97)

$$\le u_0 B + \sum_{h=1}^{H-1}4l_h(2+\psi|\mathcal{S}_h|)\big(\beta_{\bar{H}} + \tfrac{C\sqrt{l_{h-1}(2+\psi|\mathcal{S}_{h-1}|)}}{l_{h-1}\psi\lambda}\big)\sqrt{\tfrac{\eta(2\lambda+1)\gamma_T}{l_{h-1}}}$$
(98)

$$= u_0 B + \sum_{h=1}^{H-1}8(2+\psi|\mathcal{S}_h|)\big(\beta_{\bar{H}}\sqrt{\eta(2\lambda+1)l_{h-1}\gamma_T} + \tfrac{C\sqrt{(2+\psi|\mathcal{S}_{h-1}|)\eta(2\lambda+1)\gamma_T}}{\psi\lambda}\big)$$
(99)

$$\le u_0 B + \sum_{h=1}^{H-1}8(2+\psi|\mathcal{S}_h|)\big(\beta_{\bar{H}}\sqrt{\eta(2\lambda+1)T\gamma_T} + \tfrac{C\sqrt{(2+\psi|\mathcal{S}_{h-1}|)\eta(2\lambda+1)\gamma_T}}{\psi\lambda}\big)$$
(100)

$$\le u_0 B + 8\bar{H}(2+\tfrac{2\psi}{\ln\eta}\gamma_T)\big(\beta_{\bar{H}}\sqrt{\eta(2\lambda+1)T\gamma_T} + \tfrac{C\sqrt{(2+\frac{2\psi}{\ln\eta}\gamma_T)\eta(2\lambda+1)\gamma_T}}{\psi\lambda}\big),$$
(101)

where Eq. (97) follows from the bound on $u_h$ in Lemma 11, Eq. (98) from the monotonicity of $\beta_h$ in $h \in \{1, \ldots, \bar{H}\}$ and $\gamma_t \in \{1, \ldots, T\}$ in $t$ (see Lemma 10 for the statement that $h \leq \bar{H}$), Eq. (99) by rearranging and using $l_h = 2l_{h-1}$, Eq. (100) by upper bounding $l_{h-1}$ by $T$, and Eq. (101) from the bound on $|\mathcal{S}_h|$ in Lemma 14.

By setting, $\psi = \frac{\ln \eta}{2\gamma_T}$ as in the theorem statement, it follows that

$$R_T \leq u_0 B + 24\bar{H}\left(\beta_{\bar{H}}\sqrt{\eta(2\lambda+1)T\gamma_T} + C\sqrt{\frac{12\eta(2\lambda+1)\gamma_T^3}{\lambda^2(\ln\eta)^2}}\right). \tag{102}$$

Treating $\lambda > 0$ as a constant, it suffices to set the switching parameter $\eta$ to some constant value (above one), so we choose $\eta = e$ (Euler's number). Then, we note that $u_0 = O(1)$ by design in the algorithm (recall that $l_0 = 2$, and note that $\psi \leq 1$ except possibly when $T$ is small), and we write our regret bound as

$$R_T \leq O\left(\bar{H}(\beta_{\bar{H}}\sqrt{T\gamma_T} + C\gamma_T^{3/2})\right). \tag{103}$$

By using the notation $O^*(\cdot)$ to hide the multiplicative $\bar{H} = \log_2 T$ factor, the final result then follows:

$$R_T \leq O^*\left(\beta_{\bar{H}}\sqrt{T\gamma_T} + C\gamma_T^{3/2}\right). \tag{104}$$

$\square$

# E  Alternative Approach: Reduction to Linear Bandits

In this section, we introduce an alternative method for corrupted kernelized bandit optimization, and discuss its limitations. We reduce the kernelized bandit problem of dimension $d$ to a linear bandit problem of dimension $D$[3] using techniques from [44], and then solve the corrupted linear bandit problem using a modified version of the Robust Phased Elimination algorithm [9].

We consider a finite set of $D$ actions $\mathcal{X}_D = \{s_1, \ldots, s_D\} \subseteq \mathcal{X}$, and denote by $V(\mathcal{X}_D)$ the vector subspace of $\mathcal{H}_k$ spanned by $\{k(\cdot, s_i) : s_i \in \mathcal{X}_D\}$. Following [44], we consider using the orthogonal projection $\Pi_D(f)$ of $f$ onto $V(\mathcal{X}_D)$ as an approximation of $f$, where $\Pi_D(f)$ is also the unique interpolant of $f$ on $\mathcal{X}_D$ in $V(\mathcal{X}_D)$, i.e., $\Pi_D(f)(s_i) = f(s_i)$ for $i = 1, \ldots, D$. To design this set $\mathcal{X}_D$, we use Algorithm 2 (taken from [44]), which takes the kernel $k$, domain $\mathcal{X}$, and an admissible error $e$ as input, and outputs $\mathcal{X}_D$ along with the Newton basis of $V(\mathcal{X}_D)$. Recalling that $\|f\|_k \leq B$, we run Algorithm 2 with admissible error $e = \Delta/B$ for some constant $\Delta > 0$. We will discuss the choice of $\Delta$ later.

---

**Algorithm 2** Newton Basis Construction [44]

**Input:** Kernel $k$, domain $\mathcal{X}$, admissible error $e$
**Output:** $\mathcal{X}_D = \{s_1, \ldots, s_D\} \subseteq \mathcal{X}$, Newton basis $N_1, \ldots, N_D$ of $V(\mathcal{X}_D)$
 1: $s_1 \leftarrow \arg\max_{x \in \mathcal{X}} k(x, x)$
 2: $N_1(x) \leftarrow k(x, s_1)/\sqrt{k(s_1, s_1)}$
 3: **for** $D \leftarrow 1, 2, \ldots$ **do**
 4:     Define $P_D^2(x) = k(x, x) - \sum_{i=1}^{D} N_i^2(x)$
 5:     **if** $\max_{x \in \mathcal{X}} P_D^2(x) < e^2$ **then**
 6:         **return** $\{s_1, \ldots, s_D\}$ and $\{N_1, \ldots, N_D\}$
 7:     **end if**
 8:     $s_{D+1} \leftarrow \arg\max_{x \in \mathcal{X}} P_D^2(x)$
 9:     $u(x) \leftarrow k(x, s_{D+1}) - \sum_{i=1}^{D} N_i(s_{D+1})N_i(x)$
 10:     $N_{D+1}(x) \leftarrow u(x)/\sqrt{P_D^2(s_{D+1})}$
 11: **end for**

---

By rearranging the equations in Theorem 6 of [44], we have that the number of points returned by the algorithm is $D = O\left((\log \frac{1}{\Delta})^d\right)$ for kernels with infinite smoothness (in particular, the SE kernel), and $D = O(\Delta^{-d/\nu})$ for kernels with finite smoothness $\nu$ (in particular, the Matérn-$\nu$ kernel).

---

[3]The notation $D$ for the continuous domain $[0,1]^d$ will not be used in this appendix, so it it safe to use $D$ for this dimension quantity.

Since the Newton basis $\{N_1, \ldots, N_D\}$ returned is the Gram-Schmidt orthonormalization of the basis $\{k(\cdot, s_i) : s_i \in \mathcal{X}_D\}$, we have for any $f \in \mathcal{H}_k$ and $x \in \mathcal{X}$ that

$$\left| f(x) - \sum_{i=1}^{D} \langle f, N_i \rangle N_i(x) \right| \le \|f\|_k \cdot e \le \Delta \tag{105}$$

under the choice $e = \Delta/B$. Hence, for any fixed black-box $f \in \mathcal{H}_k$ with $\|f\|_k \le B$, there exists a $\theta \in \mathbb{R}^D$ with $\|\theta\|_2 \le B$ such that for any $x \in \mathcal{X}$,

$$|f(x) - \langle \theta, \widetilde{x} \rangle| \le \Delta, \tag{106}$$

where for any given point $x$, we define $\widetilde{x} = [N_1(x), \ldots, N_D(x)]^T$. Now, we can reduce the corrupted kernelized bandit problem to a variant of the corrupted linear bandit problem [9] on the transformed domain $\widetilde{\mathcal{X}} = \{\widetilde{x} : x \in \mathcal{X}\}$ of dimension $D$, where $|\widetilde{y}_t - \langle \theta, \widetilde{x}_t \rangle - c_t - \epsilon_t| \le \Delta$ for $t = 1, \ldots, T$.

## E.1 A Variant of Robust Phased Elimination

We apply Algorithm 3, a variant of the Robust Phased Elimination algorithm for stochastic linear bandits [9], on the space $\widetilde{\mathcal{X}}$ of dimension $D$, where the only difference from the original algorithm is the confidence bound in the elimination rule.

---

**Algorithm 3** Robust Phased Elimination

---

**Input:** Actions $\widetilde{\mathcal{X}} \subseteq \mathbb{R}^D$, kernel $k$, admissible error $e$, confidence $\delta \in (0, 1)$, truncation parameter $\alpha \in (0, 1)$, time horizon $T$

1: $h \leftarrow 0, m_0 \leftarrow 4D(\log \log D + 18), \mathcal{A}_0 \leftarrow \widetilde{\mathcal{X}}$.
2: Compute design $\zeta_h : \mathcal{A}_h \to [0, 1]$ such that

$$\max_{\widetilde{x} \in \mathcal{A}_h} \|\widetilde{x}\|^2_{\Gamma(\zeta_h)^{-1}} \le 2D, \text{ and } |\text{supp}(\zeta_h)| \le m_0, \tag{107}$$

   where $\Gamma(\zeta_h) = \sum_{\widetilde{x} \in \mathcal{A}_h} \zeta_h(\widetilde{x}) \widetilde{x} \widetilde{x}^T$ (e.g., using Frank-Wolfe [30])
3: $u_h(\widetilde{x}) \leftarrow 0$ if $\zeta(\widetilde{x}) = 0$, and $u_h(\widetilde{x}) \leftarrow \lceil m_h \max\{\zeta_h(\widetilde{x}), \alpha\} \rceil$ otherwise.
4: Take each action $x$ such that $\widetilde{x} \in \mathcal{A}_h$ exactly $u_h(\widetilde{x})$ times, and get rewards $\{\widetilde{y}_t\}_{t=1}^{u_h}$, where $u_h = \sum_{\widetilde{x} \in \mathcal{A}_h} u_h(\widetilde{x})$.
5: Estimate the parameter vector $\widetilde{\theta}_h$:

$$\widetilde{\theta}_h = \Gamma_h^{-1} \sum_{t=1}^{u_h} \widetilde{x}_t u_h(\widetilde{x}_t)^{-1} \sum_{s \in \mathcal{T}(\widetilde{x}_t)} \widetilde{y}_s, \tag{108}$$

   where $\Gamma_h^{-1} = \sum_{\widetilde{x} \in \mathcal{A}_h} u_h(\widetilde{x}) \widetilde{x} \widetilde{x}^T$ and $\mathcal{T}(\widetilde{x}) = \{s \in \{1, \ldots, u_h\} : \widetilde{x}_s = \widetilde{x}\}$.
6: Update the active set of actions:

$$\mathcal{A}_{h+1} \leftarrow \left\{ \widetilde{x} \in \mathcal{A}_h : \max_{\widetilde{x}' \in \mathcal{A}_h} \langle \widetilde{\theta}_h, \widetilde{x}' - \widetilde{x} \rangle \le 4\Delta\sqrt{D(1 + \alpha m_0)} \right.$$
$$\left. + 4\sqrt{\frac{D}{m_h} \log \frac{1}{\delta}} + \frac{4C}{\alpha m_h}\sqrt{D(1 + \alpha m_0)} \right\}. \tag{109}$$

7: $m_{h+1} \leftarrow 2m_h, h \leftarrow h + 1$ and return to step 3 (terminating after $T$ actions are played).

---

The analysis of Algorithm 3 is very similar to that of [9], so we heavily rely on their auxiliary results and only focus on explaining the differences here. With $\widetilde{\theta}_h$ denoting the estimate of $\theta$ based on the corrupted observations $\{\widetilde{y}_t\}_{t=1}^{u_h}$ in the algorithm, and $\widehat{\theta}_h$ denoting the estimate of $\theta$ based on $\{\langle \theta, \widetilde{x}_t \rangle + c_t + \epsilon_t\}_{t=1}^{u_h}$ (i.e., the corrupted observations if the linear model were exact) in the original algorithm, we have for all $h \ge 0$ and $\widetilde{x} \in \mathcal{A}_h$ that

$$|\langle \widetilde{x}, \widetilde{\theta}_h - \widehat{\theta}_h \rangle| \le \left| \widetilde{x}^T \Gamma_h^{-1} \sum_{t=1}^{u_h} \widetilde{x}_t \Delta \right| \le \Delta \sum_{t=1}^{u_h} |\langle \widetilde{x}, \Gamma_h^{-1} \widetilde{x}_t \rangle| \overset{(a)}{\le} \Delta \sqrt{u_h} \|\widetilde{x}\|_{\Gamma_h^{-1}} \overset{(b)}{\le} 2\Delta\sqrt{D(1 + \alpha m_0)}, \tag{110}$$

where (a) uses the definition of $\|\cdot\|_{\Gamma_h^{-1}}$ and the fact that the $\ell_1$-norm is upper bounded by the $\ell_2$-norm times the square root of the vector length, and (b) uses Lemmas 2 and 3 of [9]. Hence, in a fixed epoch $h$, we have for all $\widetilde{x} \in \mathcal{A}_h$ that

$$|\langle \widetilde{x}, \widetilde{\theta}_h - \theta \rangle| \le |\langle \widetilde{x}, \widetilde{\theta}_h - \widehat{\theta}_h \rangle| + |\langle \widetilde{x}, \widehat{\theta}_h - \theta \rangle| \tag{111}$$

$$\le 2\Delta \sqrt{D(1 + \alpha m_0)} + 2\sqrt{\frac{D}{m_h} \log \frac{1}{\delta}} + \frac{2C}{\alpha m_h} \sqrt{D(1 + \alpha m_0)}, \tag{112}$$

where the first term uses (110), and the remaining terms are obtained with probability at least $1 - 2|\mathcal{X}|\delta$ by Lemma 4 in [9].

Defining $\bar{x} = \arg\max_{\bar{x} \in \widetilde{\mathcal{X}}} \langle \theta, \bar{x} \rangle$, by a similar analysis to Section A.2 in [9], we can show that the elimination rule in (109) retains $\bar{x}$ in a given epoch with probability at least $1 - 2|\mathcal{X}|\delta$. Recalling that $x^* = \arg\max_{x \in \mathcal{X}} f(x)$, we have

$$f(x^*) = \langle \theta, \widetilde{x}^* \rangle + f(x^*) - \langle \theta, \widetilde{x}^* \rangle \le \langle \theta, \widetilde{x}^* \rangle + \Delta \le \langle \theta, \bar{x} \rangle + \Delta. \tag{113}$$

Hence, the cumulative regret can be upper bounded as follows

$$R_T = \sum_{t=1}^{T} f(x^*) - f(x_t) \le \sum_{t=1}^{T} (\langle \theta, \bar{x} \rangle + \Delta) - (\langle \theta, \widetilde{x}_t \rangle - \Delta) = \sum_{t=1}^{T} \langle \theta, \bar{x} - \widetilde{x}_t \rangle + 2\Delta T. \tag{114}$$

Again following the analysis of Section A.2 in [9], using (112) and (109), we can then show that the cumulative regret is

$$R_T = O^* \left( \Delta T \sqrt{D} + \sqrt{DT \log \frac{|\mathcal{X}|}{\delta}} + CD^{3/2} \right) \tag{115}$$

with probability at least $1 - \delta$.

## E.2 The choice of $\Delta$

The only remaining step now is to find a proper choice of $\Delta$, which is what dictates the choice of $D$ (along with the kernel). The choice of $\Delta$ can be optimized with respect to the kernel parameters, and the optimal scaling is achieved by equating the first terms in (115) with one of the other two terms (whichever is larger). We first consider the choice $\Delta = \frac{1}{\sqrt{T}}$, which equates the first two terms (up to the $\log \frac{|\mathcal{X}|}{\delta}$ factor).

With $\Delta = \frac{1}{\sqrt{T}}$, it is known from [44, Corollary 7] that Algorithm 2 results in $D = O((\log T)^d)$ for the SE kernel and $D = O(T^{\frac{d}{2\nu}})$ for the Matérn kernel. Hence, the cumulative regret of our method is upper bounded as follows:

- For the SE kernel,

$$R_T = O^* \left( \sqrt{T(\log T)^d \log \frac{|\mathcal{X}|}{\delta}} + C(\log T)^{\frac{3d}{2}} \right). \tag{116}$$

- For the Matérn kernel,

$$R_T = O^* \left( \sqrt{T^{\frac{d+2\nu}{2\nu}} \log \frac{|\mathcal{X}|}{\delta}} + CT^{\frac{3d}{4\nu}} \right). \tag{117}$$

For the Matérn kernel, we can sometimes do better by equating the first and third terms in (115), whereas for the SE kernel this is never the case. The exact optimal choice depends on how $C$ scales with respect to $T$, but to avoid unwieldy expressions, we focus here on the direct $T$ dependence in (115) so treat $C$ as a constant. Equating the first and third terms, and ignoring the $\log T$ term, we find that we should set $\Delta = 1/T^{\frac{\nu}{d+\nu}}$, which yields $D = O(T^{\frac{d}{d+\nu}})$ [44], and gives

$$R_T = O^* \left( \sqrt{T^{\frac{2d+\nu}{d+\nu}} \log \frac{|\mathcal{X}|}{\delta}} + CT^{\frac{3d}{2(d+\nu)}} \right). \tag{118}$$

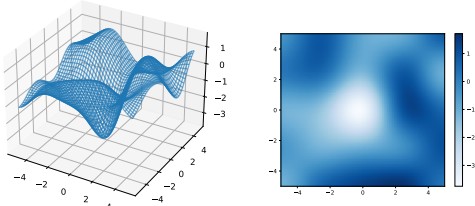

Figure 4: Illustration of 2D synthetic function.

We compare (117) and (118) for various $(\nu, d)$ pairs below.

For the SE kernel, the bound (116) turns out to be strong, matching our main result (Section 3), though we believe that our algorithm's feature of directly using the GP model (i.e., avoiding linear approximations) is still desirable.

For the Matérn kernel, however, the resulting bound is not as strong; in particular, the non-corrupted terms in both (117) and (118) are larger than the corresponding term $\sqrt{T\gamma_T} = O^*(T^{\frac{\nu+d}{2\nu+d}})$ in our main result.[4] The same goes for the corrupted terms, with the root cause for both terms being that either choice of $D$ above is strictly higher than $\gamma_T$. For the corrupted term, this is further highlighted by comparing the regimes in which the bound remains sublinear:

- The term $T^{\frac{3d}{4\nu}}$ in (117) is sublinear when $\nu > \frac{3}{4}d$;

- The term $T^{\frac{3d}{2(d+\nu)}}$ in (118) is sublinear when $\nu > \frac{d}{2}$;

- The analogous term $\gamma_T^{3/2} = O^*(T^{\frac{3d}{4\nu+2d}})$ in the main body is sublinear under the milder condition $\nu > d/4$.

Note that in general, we have for constant $C$ that (117) is a better bound than (118) when $\nu > d$, (118) is better than (117) when $\nu \in \left(\frac{d}{2}, d\right)$, and both fail to be sublinear when $\nu \le \frac{d}{2}$.

We note that a slight caveat to the preceding findings is that it is unclear whether the choice of $D$ arising from [44] is the best possible, but we are not aware of any similar results that are better for our purposes.

## F  Supplementary Experimental Results

Recall that the synthetic function $f_1$ is shown in Figure 4. This section contains the experimental results on $f_1$ with $C = 100$ (Figure 5), and on Robot3D with $C = 50$ (Figure 6). The overall findings are generally similar to those in the main text, and are not repeated here.

---

[4]For (117), this is seen by writing $T^{\frac{d+2\nu}{2\nu}} = T^{1+\frac{d}{2\nu}}$ and noting that $\frac{d}{2\nu}$ exceeds $\gamma_T = O^*(T^{\frac{d}{2\nu+d}})$. For (118), it is seen by writing $\sqrt{T^{\frac{2d+\nu}{d+\nu}}} = T^{\frac{2d+\nu}{2d+2\nu}}$, and noting that subtracting $d$ from both the numerator and denominator makes the fraction smaller.

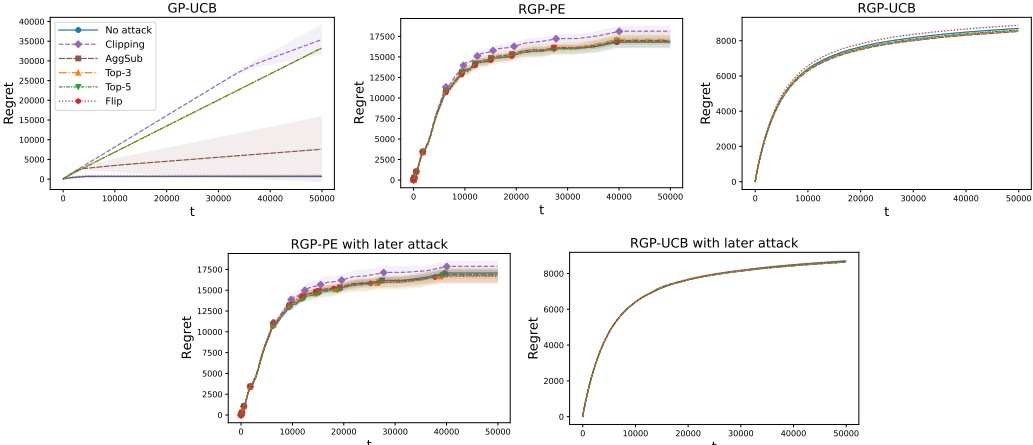

Figure 5: Performance on $f_1$ with $C = 100$. Note that for GP-UCB, the curves for Top-3 and Top-5 are indistinguishable, so only the latter is clearly visible. Similar trends are observed to the case $C = 50$ in Figure 2.

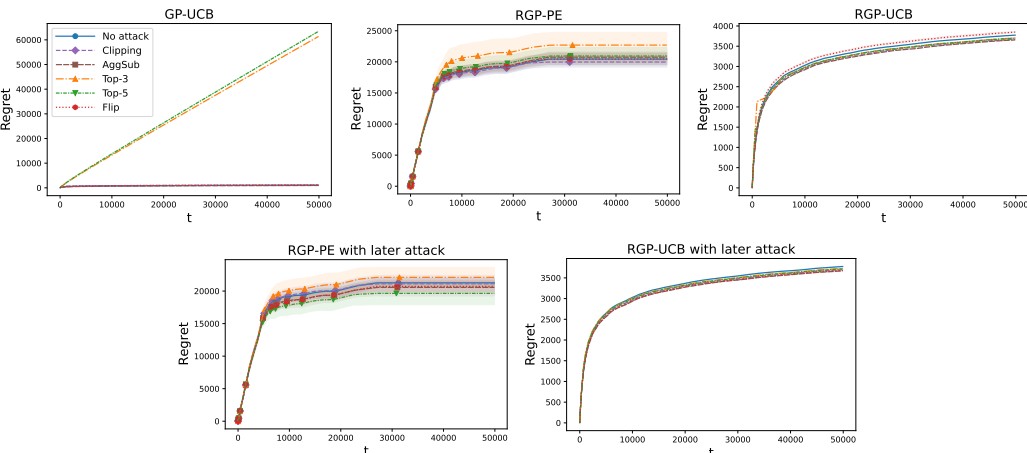

Figure 6: Performance on Robot3D with $C = 50$. Similar trends are observed to the case $C = 100$ in Figure 3.