# OpenReview forum: "A Robust Phased Elimination Algorithm for Corruption-Tolerant Gaussian Process Bandits"
_NeurIPS.cc/2022/Conference — NeurIPS 2022 Accept_

### Official Review · Reviewer_s8U7 · 2022-07-11

**Rating:** 6
**Confidence:** 4
**Soundness:** 4 excellent
**Presentation:** 4 excellent
**Contribution:** 2 fair

**Summary:**

Setting:
A sequential decision-making problem- Gaussian process bandit optimization, where the signal observed by the learner may be adversarially corrupted up to some budget parameter C.

More specifically, there is an unknown reward function f with a bounded norm in a Reproducing Kernel Hilbert Space.
In the standard model without corruptions, the learner chooses a strategy x and observes f(x) + stochastic (sub-gaussian) noise. In the model with corruption, additional adversarial noise is added to the observed signal.
The learner would like to minimize its cumulative loss over T rounds.

The main contribution is an algorithm with regret of sqrt{T*Info-gain}+C*info-gain^3/2, which improves on a previous bound of C*sqrt{T*info-gain} for some types of kernels.
The algorithm is based on confidence bounds that take into account the corruptions.
Moreover, the algorithm obtains more samples of the same point such that the average is harder to corrupt - "rare switching".



**Questions:**

I think that it is worth mentioning other papers that study very related models with corruptions, for example:
Prediction with Corrupted Expert Advice (Neurips 20),
Best-of-All-Worlds Bounds for Online Learning with Feedback Graphs (Neurips 21),
Adversarially Robust Multi-Armed Bandit Algorithm with Variance-Dependent Regret Bounds (COLT 22),
and many references therein.

**Ethics Review Area:**

["I don’t know"]

**Limitations:**

-

**Strengths And Weaknesses:**

The paper improves on a previous result from "Corruption-tolerant Gaussian process bandit optimization" (AISTATS 20).

The "rare switching" idea is nice for improving robustness.

On the negative side, C is known to the learner, as opposed to some other related models (not bandit optimization). Crucially, the corrupted confidence bounds depend on that.

---

> ### Author Response · Authors · 2022-07-30
> **Response to Reviewer s8U7**
>
> Thank you for the positive assessment of our work.
>
> Please see the separate post labeled “Common Response” regarding parameter C being known.
>
> Thank you also for the additional references.  We believe that these works are somewhat less related due to studying distinct non-kernel settings, but we will consider adding a few more citations to better emphasize that such settings have received significant attention.

---

### Official Review · Reviewer_Wgbt · 2022-07-11

**Rating:** 5
**Confidence:** 3
**Soundness:** 4 excellent
**Presentation:** 2 fair
**Contribution:** 2 fair

**Summary:**

This paper considered the bandits problem where the result function lives in an RKHS space and can be corrupted within C budgets. When C is known in advance, the author proposed a robust elimination-type algorithm that achieves a regret bound with the C term only appearing in additive terms, which significantly improves the current result where the C appears multiplicatively.   This result is near-optimal in the SE kernel. Besides the theoretical results, the authors also provide simulation results with a different type of attack.

**Questions:**

Can you give more explanation on the intuition behind the result of Lemma 2? It is not very clear to me. From its proofs in the appendix, it seems to me just a direct but more careful calculation result with corruption presented. So I am just curious why no one achieve that before. Are there any particular techniques I need to pay attention to?

**Strengths And Weaknesses:**

Strength:

1\ The author gives the first near-optimal result (in some special cases) in RKHS setting with sound proofs.

2\ The author gives refined enlarged confidence bound analysis compared to C in [7], which is somehow interesting.

3\ The authors provide simulation results for various attacks.

Shortcoming：

1) This rare switch technique, although claimed by the authors that “To our knowledge, we are the first to use rare switching to achieve adversarial robustness; previous works instead used it for reducing computational complexity.” I personally think it has been used in many other places to get stability in the algorithm analysis. Of course not totally the same, but not that novel. Especially, the author states “we obtain more samples of the same point, allowing us to average more of them together and making the “averaged” observation harder to corrupt.” But to me, it is just a variant of basic doubling techniques in most elimination-type algorithms.

Maybe there is some difficulty/novelty in incorporating “rare switch” technique in this particular framework that I do not correctly understand? If the author can illustrate more on this I am willing to adjust my score.

2)  I am not totally convinced by the author’s claim that the Gaussian process with KNOWN corruptions is a hard problem because this algorithm is mainly improved upon on the framework of [7][8] with some well-known techniques (rare-switch, enlarged confidence). Giving a complete result with unknown corruption may make the paper stronger.

---

> ### Author Response · Authors · 2022-07-30
> **Response to Reviewer Wgbt**
>
> We thank the reviewer for the overall positive assessment of our paper.
>
> (on switching) We made the claim regarding rare switching based on the best of our understanding of the existing literature, but our understanding may have been imperfect.  If the reviewer provides relevant papers where rare switching has been used for stability, we would be happy to revise the discussion accordingly (if we are in agreement).  Similarly, any specific references for doubling techniques in elimination with adversarial corruption would be appreciated.
>
> (on analysis) We believe that the parts of our analysis that are relevant to rare switching are far from trivial, and do not follow readily from existing works.  For instance, Eq. (26)-(40) includes a useful reformulation of the posterior mean in the presence of repeated samples, Lemma 12 (spanning around 1.5 pages including its proof) provides a novel result relating the switching strategy to the Gaussian Process posterior variance, and the proof of Lemma 13 is also based on the use of rare switching. We also introduced a novel, more robust, kernel mean estimator.
>
> Please see the separate post labeled “Common Response” regarding the issue of C being known.
>
> Regarding Lemma 2, perhaps the main distinction from previous work (e.g., [7]) is the inclusion of $u_{\min}$ (lower bounding how many times we repeatedly sample each point), which allows us to reduce the coefficient to $C$ by a factor of $\frac{u_h}{u_{\min}} \ll 1$ as discussed just after the lemma.  We also make use of a distinct robust mean estimator $\tilde{\mu}$, requiring a fair bit of distinct analysis in equations (26)-(41).
>
> It is difficult to confidently comment on why this hasn’t been done before, but we would certainly view this as a positive point for our work.  One reason that these ideas were not used in certain other settings is that they were simply not necessary, e.g., for corrupted linear bandits, [8] gave near-optimal regret bounds without any repeated sampling or rare switching.  Apart from that, we expect that our precise combination of techniques/ideas (and their detailed mathematical analysis) would not be obvious to most researchers that have studied this problem.

---

> > ### Comment · Reviewer_Wgbt · 2022-08-08
> > **About rare switching techniques**
> >
> > I believe this rare switch technique has been used back in 2011 [1]. And it has also been used recently, like [2] and  [3]. I admitted that the recent usage for the rare switches is more from a computational perspective instead of a regret perspective, but I personally think these two don't have such fundamental differences and it is very natural for people to control stability by dividing algorithms into several epochs. For example, in [4], I believe it is also a conceptually rare switch. Overall, I agree that adapting the rare switch idea in this setting is not trivial, but I still think it is relatively incremental.
> >
> > I will not raise my score because I personally think the technical contribution is incremental. But I admit that the result is improved and the analysis is solid. So I am ok with acceptance given the good result.
> >
> >
> > [1] Regret Bounds for the Adaptive Control of Linear Quadratic Systems http://proceedings.mlr.press/v19/abbasi-yadkori11a/abbasi-yadkori11a.pdf
> > [2] Provably Efficient Reinforcement Learning with Linear Function Approximation under Adaptivity Constraint https://arxiv.org/pdf/2101.02195.pdf
> > [3] A Provably Efficient Algorithm for Linear Markov Decision Process with Low Switching Cost https://arxiv.org/pdf/2101.00494.pdf.
> > [4] Better Algorithms for Stochastic Bandits with Adversarial Corruptions http://proceedings.mlr.press/v99/gupta19a/gupta19a.pdf

---

> > > ### Author Response · Authors · 2022-08-09
> > > **Response to about rare switching techniques**
> > >
> > > We thank the reviewer for acknowledging the novelty of our results, solid analysis, and being ok with acceptance. We will further elaborate on the rare switching technique (+ add some of the missing references).

---

### Official Review · Reviewer_ZcbT · 2022-07-17

**Rating:** 6
**Confidence:** 3
**Soundness:** 3 good
**Presentation:** 3 good
**Contribution:** 2 fair

**Summary:**

The paper proposes a new algorithm for robust optimization of Gaussian process bandits and prove that it provides tighter regret guarantees with respect to the corruption budget (in an additive model of reward corruptions). The algorithm divides the time horizon into phases of increasing length, where in each phase the action set is pruned and only a limited number of actions (based on an information gain criterion) are actually played. The proofs and the empirical proofs both show that the algorithm is robust to corruptions.

**Questions:**

1. It would be useful to understand the performance of the algorithm in a setting where the corruption budget $C$ is misspecified, and compare it with the other robust GP algorithm [1].
1. What is the optimal adversarial strategy for allocating corruption budget? Including it as one of the attack methods in Section 4, if feasible, would be useful.
1. Minor: please define before you use SE (line 62) and $GP(0,k)$ (line 99) .

[1] http://proceedings.mlr.press/v108/bogunovic20a/bogunovic20a.pdf

**Limitations:**

Not applicable.

**Strengths And Weaknesses:**

Strengths:
- The paper is very well-written -- the setting, comparison with related work, the algorithm and their main contributions are all clearly identified.
- The work covers significant ground, such as, working through older techniques that do not readily lead to the tighter bounds provable with the proposed algorithm.

Limitations:
- The proposed algorithm assumes that the corruption budget $C$ of the adversary is known in advance (that is, knowledge of $C$ is needed to operationalize the algorithm itself, not just to prove its regret bounds). It is not clear how to set this practically, or what the effect of misspecification might be.
- The algorithm introduces significant complexity which may not be justified by the benefits it brings. The empirical results show that the regret scales similar to another robust GP algorithm [1] (and the performance is in fact worse in absolute terms). It leaves the open the question of whether a tighter bound can be obtained for the previous approach. As such, it is not clear if there exists a setting where this algorithm shines and how important the corruption term itself is.
- Minor: the results hold under fairly strong assumptions on the model (e.g. variance in eq (6) unaffected by corruptions by design, bounded rewards which satisfy budget constraints under addition)

[1] http://proceedings.mlr.press/v108/bogunovic20a/bogunovic20a.pdf

---

> ### Author Response · Authors · 2022-07-30
> **Response to Reviewer ZcbT**
>
> We thank the reviewer for the overall positive assessment of our paper.
>
> Please see the separate post labeled “Common Response” regarding the issue of C being known.
>
> (on complexity) We do not believe our algorithm to be significantly more complex than the previous ones, and yet it achieves improved results as summarized in Table 1.  From a theoretical viewpoint, this is an immediate and clear benefit.  From a practical viewpoint, we implemented our algorithm without any major difficulties and used it in our experiments.
>
> (on alg. from  [1]) The experiments do show that an algorithm from [1] can be better in practice, but we believe we were quite open about this and adequately highlighted the fact that improved theory for that algorithm could be possible.  But without there existing any such theory (which could be very difficult or even impossible), we do not believe that the mere *possibility* of this should be viewed as a negative point for this paper.
>
> (opt. attack strategy) We are not aware of any prior works figuring out what the optimal attack strategy is, even in simpler settings such as linear bandits.  Hence, as far as we understand, it is not currently feasible to implement such an attack experimentally.

---

### Official Review · Reviewer_objk · 2022-07-22

**Rating:** 3
**Confidence:** 4
**Soundness:** 3 good
**Presentation:** 3 good
**Contribution:** 1 poor

**Summary:**

This work focuses on Gaussian Process Bandits setting where the reward observations are corrupted by an adversary. The work proposes an algorithm robust to adversarial corruption. The work shows that the regret of the algorithm in O(C\gamma^{3/2}_T), where C is the amount of corruption, T is the time horizon, and \gamma_{T} is the maximal information gain. This bound is tighter than the previously existing bound O(C\sqrt{T\gamma_{T}}).

The key difference between the proposed algorithm and the previous algorithm in this setting is that the proposed algorithm uses the trick of phased elimination based on confidence bounds to achieve this additive dependence on C (instead of multiplicative dependence). The analysis is carried out under an additional assumption, Assumption 1,  on the non-corrupted observations.


**Questions:**

1)	I would suggest authors to clarify the motivation behind Assumption 1. Is it common in the literature? If yes, please point to the references? If not, please clarify the comparison of the theoretical results with the literature.

2)	Can you please provide intuition regarding the Assumption 1 being applicable to a specific estimator? Is there an analogue of this assumption in terms of assumption on the setting instead of a specific estimator?

3)	How does the strategy in Lemma 4 applicable to the proposed algorithm under Assumption 1 given the independence assumption?


**Limitations:**

Yes

**Strengths And Weaknesses:**

Originality:
The novelty of the algorithm is limited. A phased elimination trick is applied to the previously known UCB estimator of parameters in Gaussian Process Bandits setting. In literature, this strategy of Phased elimination has been applied in linear bandits to achieve a robust phased elimination algorithm. Since Gaussian Process Bandits setting has been studied previously, the proposed algorithm is a combination of these works.

Additionally, the technical analysis is carried out under Assumption 1. This assumption is not common in the previous works in the literature. The need for this assumption is unclear since it is applied over an estimator used in a setting without corruption.

The work presents a few examples of sampling strategies that satisfy Assumption 1. However, in Lemma 4, the assumption states that the sample x_t is independent of all y_i, i<t. In an online setting, this assumption may not hold since sample x_t is dependent on the previously collected samples.

Quality and Clarity: Paper is well-written and key ideas developed in the paper are discussed.

Significance:

The significance of the setting and the paper is unclear because the results are not directly comparable with the literature, and the novelty in the algorithm is limited.

---

> ### Author Response · Authors · 2022-07-30
> **Response to Reviewer objk**
>
> > Originality: The novelty of the algorithm is limited. A phased elimination trick is applied to the previously known UCB estimator of parameters in Gaussian Process Bandits setting. In literature, this strategy of Phased elimination has been applied in linear bandits to achieve a robust phased elimination algorithm. Since Gaussian Process Bandits setting has been studied previously, the proposed algorithm is a combination of these works.
>
> Our algorithm achieves novel results in comparison to the previous corrupted GP bandit works (Table 1) and uses different algorithmic techniques. A direct reduction to the phased elimination strategy as used in the linear bandit case leads to worse regret bounds than ours, as elaborated in detail in Appendix E.  Basically, this appendix serves to show that direct reduction to existing techniques is *not* sufficient.
>
> We strongly disagree with the notion that using phased elimination implies limited novelty – the vast majority of bandit algorithms are variations of UCB, Thompson sampling, phased elimination, and many such variations are nevertheless highly novel.  Our algorithm and analysis are not merely a combination of existing works – we needed to carefully design the robust estimator, the rare switching rule, the enlarged confidence bounds, etc., and our mathematical analysis in Appendices B to D bears minimal overlap with existing works.
>
> In view of the above, we do not believe the novelty of our algorithm and results to be limited.
>
> > Additionally, the technical analysis is carried out under Assumption 1. This assumption is not common in the previous works in the literature. The need for this assumption is unclear since it is applied over an estimator used in a setting without corruption.
>
> Assumption 1 simply states that some generic confidence bound holds; upon substituting any specific choice (e.g., from prior works such as [14,37,39]), this moves from an “assumption” to a “fact”.  Such specific substitutions are done in Section 3.2.  Basically, all we have done is presented things a bit differently to prior works (first generic, then specific), but the assumption poses no additional limitations at all compared to other works.
>
> The reviewer is correct that this assumption is made for the non-corrupted setting.  This is not inconsistent with the fact that our setting has corruptions, because our analysis uses the non-corrupted confidence bound as a stepping stone to the corrupted confidence bound.  Specifically, in Eq. (52), we upper bound the error *with corruption* by $ | \tilde{\mu}_t(x) - f(x)|  \leq  | \mu_t(x) - f(x) | + | \tilde{\mu}_t(x) - \mu_t(x) | $  – then the first term amounts to a non-corrupted bound (so we can apply Assumption 1), and the second term requires a separate treatment.
>
>
> > The work presents a few examples of sampling strategies that satisfy Assumption 1. However, in Lemma 4, the assumption states that the sample x_t is independent of all y_i, i<t. In an online setting, this assumption may not hold since sample x_t is dependent on the previously collected samples.
>
> It is correct to say that *general* online algorithms could not make use of Lemma 4.  However, our query strategy in every epoch does not depend on the previously obtained observations (see Line 4 in Algorithm 1). Moreover, we apply Lemma 4 for every epoch *separately*, and in Line 14 we only use the observations from the current epoch to compute the robust estimator. Hence, our application of Lemma 4 is perfectly valid, as our algorithm was specifically designed to allow it.
>
> Since the reviewer’s rating is currently described as “...a paper with technical flaws, weak evaluation, inadequate reproducibility…”, we kindly ask that they reconsider this in view of our response.  (Alternatively, we would appreciate an explanation if the reviewer still believes that there is a flaw.)
>
> > The significance of the setting and the paper is unclear because the results are not directly comparable with the literature, and the novelty in the algorithm is limited.
>
> We have an entire subsection (Section 3.3) devoted to directly comparing our results to the existing literature, as well as providing a summary in Table 1, and experimental comparisons in Section 4.  Accordingly, we respectfully disagree with this comment, and kindly ask the reviewer to clarify/elaborate on this if they are not convinced by our response.
>
> We have answered all of the questions provided by the reviewer and we have disputed the claim of our analysis being flawed, and we firmly believe that none of these issues amount to any significant changes being required in the paper.  Please feel free to let us know if you have any further questions or concerns.

---

> > ### Author Response · Authors · 2022-08-09
> > **Request for acknowledgment of our responses**
> >
> > We are grateful for your feedback. We kindly request your acknowledgment of our responses, and that you let us know if there are any issues that you still find problematic, and/or check that your score is in agreement with your updated understanding of our work.

---

### Author Response · Authors · 2022-07-30
**Common response**

We are grateful to all of the reviewers for their feedback and suggestions.  We will respond to most points in separate replies to each reviewer, but since some reviewers mentioned the issue of C being known, we will provide a common response to that issue in this separate post.

Firstly, we expect unknown-C extensions to be possible in a similar spirit to [8], but since the known C case is already challenging, and brings novel regret scalings in comparison to the previous works, we prefer not to obfuscate our new ideas with the added technical difficulty of addressing unknown C.

Secondly, it is worth noting that only an *upper bound* on C is required, not an exact value.  For example, in Corollary 5, if we set $C = O( \sqrt(T) / \gamma_T )$, then the second term becomes order-wise the same as the first.  This means that for any corruption level up to this value, we get the same order-wise regret as the uncorrupted setting.  For highly smooth kernels such as linear and SE, this C value is nearly as high as $\sqrt{T}$.  This may not seem as high as ideal, but at least in the special case of linear bandits, it is the best we can hope for unless the algorithm attains significantly higher uncorrupted regret; see Theorem 4 and Section 2.2 in [8].  In view of this discussion, we believe that our results do have value even when C  is not known precisely, though we acknowledge that more work would be needed to obtain a more complete understanding.

In view of the above, we believe that our study of the known C setting is of significant value to the community.

---

### Author Response · Authors · 2022-08-08
**Author-Reviewer discussion**

Dear reviewers,

The author-reviewer discussion period is ending in less than two days.   We kindly request your acknowledgment of our responses, and that you let us know if there are any issues that you still find problematic, and/or check that your score is in agreement with your updated understanding of our work.  We particularly highlight the following:
- **Reviewer objk:** We have responded to all of your questions/concerns, and disputed the claim of our analysis being flawed.  We firmly believe that none of these issues amount to any significant changes being required in the paper.  We kindly ask that you reconsider your rating of Contribution (1, the lowest possible) and Overall Score (3, indicating major technical flaws or weak evaluation).
- **Reviewer Wgbt:** Since your review indicated that you may be willing to adjust your score, we kindly ask you to indicate your decision on whether to do so.
- **All reviewers:** Please see our "Common response" thread regarding the issue of unknown C

Thank you for your time and consideration.

---

### Meta-Review · Area_Chair_MGCu · 2022-08-28

**Recommendation:** Accept
**Confidence:** Less certain

**Metareview:**

The paper studies Gaussian process bandit optimization in the adversarial corruption model. This setting was considered in the work of [7] and regret bounds were presented where the adversarial term contains both the time horizon T and the corruption level C, which is not ideal. The current paper presents an improved algorithm that for certain kernels such as SE removes the dependence on T in the adversarial term and for other settings such as linear recovers the current best results.

The reviewers likes the contributions of the paper. One of the reviewers (objk) raised an objection about one of the assumptions mentioned in the paper and the authors clarified that the assumption is just a condition that can always be satisfied by appropriate choices of parameters. While the reviewer was not convinced, based on my own reading of the paper, I side with the authors.

This is a decent theoretical contribution. The one weakness of the work is that it assumes that the corruption level (C) is known in advance (or an upper bound on it). Prior works on GP bandit optimization and also in the case of multi-armed bandit settings handle unknown corruptions. As a result this is a borderline paper but I'm slightly leaning towards acceptance at this time.

**Award:**

No

---

### Decision · Program_Chairs · 2022-09-14

Accept